# Body size and life history shape the historical biogeography of tetrapods

Sarah-Sophie Weil [1,2] ✉, Laure Gallien [1,4], Michaël P. J. Nicolaï [3], Sébastien Lavergne[1], Luca Börger [2] & William L. Allen [2,4]

Dispersal across biogeographic barriers is a key process determining global patterns of biodiversity as it allows lineages to colonize and diversify in new realms. Here we demonstrate that past biogeographic dispersal events often depended on species' traits, by analysing 7,009 tetrapod species in 56 clades. Biogeographic models incorporating body size or life history accrued more statistical support than trait-independent models in 91% of clades. In these clades, dispersal rates increased by 28–32% for lineages with traits favouring successful biogeographic dispersal. Differences between clades in the effect magnitude of life history on dispersal rates are linked to the strength and type of biogeographic barriers and intra-clade trait variability. In many cases, large body sizes and fast life histories facilitate dispersal success. However, species with small bodies and/or slow life histories, or those with average traits, have an advantage in a minority of clades. Body size–dispersal relationships were related to a clade's average body size and life history strategy. These results provide important new insight into how traits have shaped the historical biogeography of tetrapod lineages and may impact present-day and future biogeographic dispersal.

The rare occasions in species' evolutionary histories when populations successfully disperse across major geographic barriers, such as oceans, mountain ranges or deserts, can have major consequences for the distribution of life on Earth. For example, long-distance dispersal from Africa to South America led to the evolution of over 90 species of New World monkeys[1,2], and a few chameleons rafting on vegetation from Africa to Madagascar is why today half of all living chameleon species are found in Madagascar[3,4]. However, we still know little about the determinants of these biogeographic dispersal events.

While chance has a large role to play, species' traits could also influence outcomes of biogeographic dispersal events[5]. Species' traits are known to mediate active, short-distance dispersal in animals[6,7]; however, traits are not necessarily related to biogeographic dispersal in the same manner as the mode of dispersal differs. Short-distance dispersal is primarily determined by species' active movements, whereas in biogeographic dispersal (for example, in trans-oceanic dispersal) passive transportation plays a bigger role[5]. Several traits may be related

to biogeographic dispersal success in animals, and body size and life history are likely to be crucial: body size determines relative energy requirements[8] and hence resistance to stress, such as water and food shortage. Indeed, recent findings show that large-bodied species have crossed biogeographic barriers more often than small ones in three reptile clades[9–11]. Life history strategy, defined by the trade-offs between traits related to growth, reproduction and survival[12], can influence the likelihood of populations establishing in new locations. Species with a fast life history strategy reproduce quickly, which makes founder populations more resistant to stochastic extinction[13–15]. Species with a slow life history strategy, on the other hand, exhibit less demographic variability, which makes their populations more resistant to environmental stochasticity[16–18]. Both effects are documented in chameleons, for which successful biogeographic dispersal is associated with both extremely fast and extremely slow life history strategies[11]. However, we do not know whether these initial findings in three small reptile groups represent a general pattern of trait–dispersal relationships

[1]CNRS, Laboratoire d'Ecologie Alpine, University Savoie Mont Blanc, University Grenoble Alpes, Grenoble, France. [2]Department of Biosciences, Swansea University, Swansea, UK. [3]Biology Department, Evolution and Optics of Nanostructures Group, Ghent University, Ghent, Belgium. [4]These authors jointly supervised this work: Laure Gallien, William L. Allen. ✉e-mail: sarah-sophie.weil@gmx.de

in tetrapods, or whether multiple relationships exist across highly different clades.

In this Article, we use trait-dependent models of historical biogeography at a global scale to fill this gap. We aim to understand whether species' traits (body size and life history strategy, inferred through phylogenetic factor analysis (PFA)) facilitate or hinder dispersal across major biogeographic barriers. We thus focus on lineage dispersal (sensu Hackel and Sanmartín[19]), which includes both successful movement or transport to a new biogeographic region and establishment there. We investigate whether and how trait–dispersal relationships vary across tetrapods and test how observed patterns relate to potential ecological drivers to examine mechanisms underlying any differences between clades. Understanding whether and how traits determine biogeographic dispersal outcomes, and what explains variation in trait–dispersal relationships between clades, gives insight into the history of life on Earth[20] and how species might respond to future global changes[21].

## Results and discussion
### Traits in biogeographic models
First, we investigated the importance of traits in the biogeographic histories of 56 tetrapod clades for which sufficient data are available to address our research questions. We compiled phylogenetic data, extant species' distributions and trait data for 7,009 species (on average 125 species per clade, minimum: 32 species, maximum: 491 species) spread across 10 amphibian clades, 15 mammal clades, 17 reptile clades and 14 bird clades (Fig. 1, Supplementary Table 1 and Methods). To determine the two traits of interest (body size and life history strategy), we used a PFA[22,23] to position species along two main latent factors of trait variation per clade. The first factor represented body size and related life history trait covariation, and the second factor, body size-independent life history covariation. For most clades, the life history factor was consistent with a fast–slow life history continuum, mainly determined by clutch or litter size (Extended Data Fig. 1). We then determined the relationship between species' positions on these two trait factors and their past dispersal rates, developing further the methodology of Weil et al.[11]. Briefly, we defined informative biogeographic regions for each clade using a data-driven approach (after Holt et al.[24] and Kreft and Jetz[25]) and used these biogeographic regions to estimate ancestral range changes along the phylogeny. We employed both biogeographic models where dispersal rates were independent of species' traits (trait-independent models) and models where dispersal rates depended on traits (trait-dependent models)[26]. We included distance-independent and distance-dependent extensions of each model, where in the latter, dispersal probabilities decrease with increasing distance between biogeographic regions (for details, see Weil et al.[11]). To assess whether the chosen traits have played a role in clades' biogeographic histories, we compared the performance of trait-dependent and trait-independent models using their model weights based on the corrected Akaike Information Criterion (AICc). We studied trait–dispersal relationships across the trait spectrum in binary trait-dependent models by analysing four binary splits of the continuous traits (for details, see Methods).

Our results show that the inclusion of traits generally improves models of historical biogeography, providing the strongest evidence at this scale that biological differences between lineages impact biogeographic movements[20]. In 91% of clades, the best trait-dependent model accrued more than 50% of the AICc weight (body size: 40 clades, 71%; life history: 48 clades, 86%). Even when accounting for multiple binarization thresholds by averaging trait-dependent AICc weights per clade, trait-dependent models were better supported than trait-independent models in 66% of clades (Fig. 2c).

To understand the consequences of including traits in estimations of biogeographic histories of clades, we evaluated three metrics: (1) ancestral range resolution (measured as the proportion of the most likely ancestral range at each node), (2) the average number of dispersal events per clade and (3) the proportion of node ranges that were estimated differently between trait-dependent and trait-independent models. To benchmark this comparison, we included distance-dependent and distance-independent model extensions in this assessment (for details, see Methods). There were no significant differences in the resolution of ancestral range estimations between trait-dependent, trait-independent, distance-dependent and distance-independent models (using mixed effect models, $\chi^2(3) = 7.27$, $P = 0.064$). The number of estimated dispersal events per clade did not differ significantly between trait-dependent and trait-independent models (contrasts among estimated marginal means, $t(389) = 1.53$, $P = 0.42$), but including distance between regions increased the number of dispersal events by ca. 2% ($t(389) = 7.38$, $P < 0.001$). The most likely ancestral ranges differed in 6% of phylogeny nodes on average, both between trait-dependent and trait-independent estimations, and between distance-dependent and distance-independent models.

In summary, including traits in biogeographic models changes neither the resolution of ancestral range estimations, nor does it change the number of dispersal events estimated on the phylogeny. However, in a small number of nodes (6% on average), the identity of ancestral ranges differs between trait-dependent and trait-independent estimations. These differences could be due to a change in the dispersal path and/or the timing of dispersal events. If the timing of dispersal events changes due to the inclusion of traits, then this may impact estimations of speciation modes (for example, dispersal of the ancestor followed by sympatric speciation, or two separate dispersal events followed by founder-event speciation), with implications for our understanding of different speciation processes.

### The effect of traits on dispersal rates
Incorporating body size and life history traits resulted in better biogeographic models in 91% of tetrapod clades studied (that is, the best trait-dependent model accrued more than 50% of the AICc weight). However, macro-evolutionary models with more free parameters have been shown to be more likely to be selected by model comparison, purely due to their complexity and insufficient models of comparison, and to lead to misleading parameter estimates[27]. A solution for this problem consists in combining parameter estimates of several models, weighting them by their likelihoods penalized by the number of parameters of each model (for example, using AICc)[28]. For each clade, we thus combined the dispersal parameters of all models via AICc-weighted averages to understand the role of body size and life history in dispersal. This way, we next investigated the magnitude of body size and life history effects within each clade, that is, the difference in dispersal rates between most-dispersive and least-dispersive lineages.

On average across all 56 clades, including body size led to differences in dispersal rates of 28% (standard deviation (s.d.) 19%) between most-dispersive and least-dispersive lineages, and including life history led to differences of 32% (s.d. 24%) in the 47/56 clades where life history was consistent with a fast–slow continuum (Extended Data Fig. 1). This corresponds to a median increase in dispersal rates of lineages with disperser traits of ca. 0.02 dispersal events per million years (s.d. body size 0.06, life history 0.13), compared with dispersal rates of lineages with non-disperser traits. The magnitude of body size effects within clades differed significantly between tetrapod classes ($F(3,52) = 3.03$, $P = 0.04$; Fig. 2a). The largest effects were detected in mammals and reptiles, and the smallest ones in amphibians and birds (contrast among estimated marginal means: $t(52) = 3.00$, $P = 0.004$). Stevens et al.[7] found high phylogenetic signal in dispersal distances of aerial dispersers, which might explain the small body size effects in bird clades: high phylogenetic signal indicates little intra-clade variability, which might lead to small trait effects in our intra-clade analyses. On the other hand, there were no significant differences between tetrapod classes in the magnitude of life history effects on dispersal ($F(3,52) = 0.31$, $P = 0.82$; Fig. 2a).

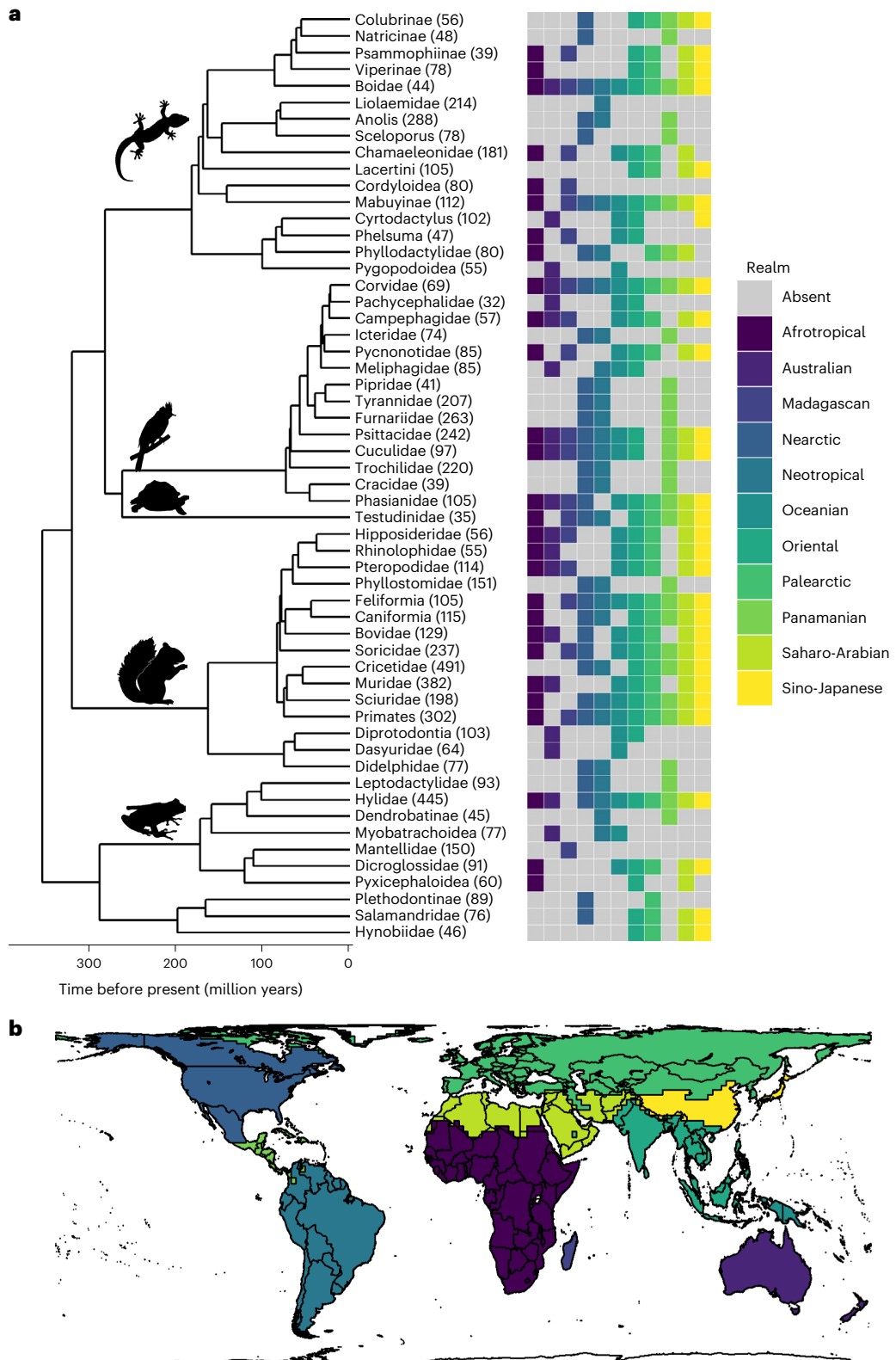

**Fig. 1 | Phylogenetic and geographic extent of the analysis for 10 amphibian, 15 mammal, 14 bird and 17 reptile clades. a,b**, The phylogenetic relationships between clades (**a**) and their occurrences in major biogeographic realms (**b**) (as determined by Holt et al.[24]). The number of species included per clade is given in parentheses after clade names.

To better understand the variation in magnitude of trait effects between clades, we investigated the influence of eight ecological and methodological variables and their associated hypotheses (Table 1): (1) the number of dispersal events per lineage as a proxy of barrier strength, (2) the proportion of estimated dispersal events that are oceanic, (3) the variability of traits within clades, (4) the number of species within a clade, (5) trait data coverage, (6) average node resolution across the phylogeny, (7) clades' crown age and (8) the

biogeographic base model (Dispersal-Extinction-Cladogenesis model (DEC), BAYAREA or DIVA models; Methods). We used a stepwise selection process on multivariate regressions to identify the variables that could best explain the maximal difference in dispersal rates between trait states across clades. Due to the limited sample size, we did not include all variables at once in a single model. Rather, we first tested the methodological variables (variables 4–8) and retained only those which were selected in the step-selection process for inclusion in a model with the three ecological variables (variables 1–3).

For body size–dispersal relationships, none of the tested variables could explain significant variation in dispersal rate differences between trait states across clades (Table 1 and Extended Data Table 1). However, for life history–dispersal relationships, we found that the difference in dispersal rates between trait states increases in clades with (1) an intermediate number of dispersal events, (2) smaller and greater proportions of oceanic dispersal events (that is, clades with mostly oceanic dispersal or mostly continental dispersal; Extended Data Fig. 2) and (3) lower life history variability (Fig. 3a). These results suggest that life history traits played a larger role in the past dispersal of clades that had to cross more stringent barriers, that is, clades where fewer dispersal events were estimated on the phylogeny, and that weak barriers may dilute the signal in trait–dispersal relationships. However, effects of life history were also smaller in clades that had to cross extremely strong barriers, which may be related to low statistical power due to the low number of dispersal events estimated on the phylogeny. Further, our results suggest that oceanic and continental barriers filter species differently, as differences in dispersal rates between trait states were more marked in clades with mostly oceanic dispersal or mostly continental dispersal than for clades with mixed oceanic and continental dispersal events. Different life history traits may therefore be related to successful dispersal in both processes, and mixing both continental and oceanic barriers can obscure life history–dispersal relationships. Biogeographic dispersal may hence be context dependent (depending on the type of barrier and possibly their characteristics, that is, environmental harshness and degree of landscape fragmentation, as has been shown for active, short-distance dispersal previously[29,30]). Future work should therefore aim to consider different types of barriers separately (which is not possible in the present study due to computational constraints; Methods and Supplementary Information Section 2). That both the number of dispersal events in a clade's biogeographic history and the proportion of oceanic dispersal events were selected in the best model also underlines the importance of choosing meaningful barriers for biogeographic models, which are specific to each clade, using, for example, data-driven bioregionalization approaches[25,31]. Furthermore, we found a negative effect of trait variability on the differences in maximal dispersal rates between clades, which is surprising and may indicate that other traits additionally influence dispersal success in clades with high life history variability (for example, habitat, diet or climatic tolerance[7,21]). Variables accounting for potential methodological biases were not significant, suggesting that they had a negligible impact on the outcome of our analyses. Our results remained qualitatively similar when we restricted the analysis to those clades that showed the strongest fast–slow life history continuum (Extended Data Table 1).

In summary, we found strong variation in the magnitude of trait effects on dispersal rates between clades. The variation in the magnitude of body size effects could not be explained by any of the variables we tested. The variation in the magnitude of life history effects, on the other hand, was related to intra-clade trait variability, as well as the strength and type of biogeographic barriers, indicating that different barriers filter species differently according to their life history strategy.

## The traits of successful dispersers

To identify the traits of successful dispersers, we further determined the shape of the relationships between traits (body size and life history) and dispersal rates in all clades where the difference in estimated dispersal rates between trait states was notable, that is, greater than 10% (79% of all body size–dispersal relationships, and 83% of the 47 life history–dispersal relationships in which fast–slow consistent trade-offs structured life history). We distinguished between the following relationships: (1) positive (large/fast-lived species were better dispersers than small/slow-lived species), (2) negative (small/slow-lived species were the better dispersers), (3) U shaped (species with extreme traits were better dispersers than intermediate ones) and (4) bell shaped (species with intermediate traits were the better dispersers).

Within clades, large-bodied species were generally better dispersers than small species (55% of all notable body size–dispersal relationships were positive), especially in reptile (73%) and mammal (54%) clades (Fig. 2b). This result is in accordance with previous findings of a large body size dispersal advantage in dispersal in general[32], and in historic biogeographic dispersal in particular[9–11]. Nevertheless, intermediate-sized species were better dispersers in 18% of clades, and small body size or extreme body sizes provided a dispersal advantage in 14% of clades. There were significant differences in the distribution of body size–dispersal relationships between ectotherms and endotherms (Fisher's exact test, $P = 0.02$); notably, there were no amphibian or reptile clades in which only small species had a dispersal advantage. Our results contrast with the findings of Stevens et al.[7], who found a positive body size–dispersal relationship in rodents (in our study: one-third of rodent clades) and a U-shaped one in passerines (in our study: three of nine passerine clades). This indicates that body size–dispersal relationships depend on the type and stage of dispersal considered (Stevens et al.[7] studied active, natal dispersal at the movement stage, while we studied successful biogeographic dispersal).

Life history–dispersal relationships were more varied in their shape than body size–dispersal relationships (Fig. 3b). Within clades, fast species were generally better dispersers than slow species (44% of the 39 clades with fast–slow consistent life history and notable dispersal relationships). However, species with an extreme life history strategy (that is, either fast or slow) were better dispersers than those with an intermediate strategy in 23% of clades, slow species were better dispersers than fast species in 18% of clades and intermediate

---

**Fig. 2 | The role of traits in biogeographic dispersal. a**, Distributions of maximal differences of dispersal rates between most-dispersive and least-dispersive states per tetrapod class, in blue for body size–dispersal relationships, in yellow for life history–dispersal relationships. Centre line, median; box limits, first and third quartile; maximum extent of whiskers, box limits ±1.5 × interquartile range (that is, the distance between the first and third quartile). **b**, Proportions of different trait–dispersal relationships per tetrapod class, including only clades where maximal differences in dispersal rates between trait states were greater than 10%. 'Extreme body size/life history' refers to clades where a U-shaped relationship was inferred between traits and dispersal rates, that is, where species with extremely small or large body sizes, or fast or slow life histories had a dispersal advantage. **c**, Relative AICc weight of trait-dependent models compared with the entire set of candidate models, and across the entire range of binary trait thresholds (that is, the sum of AICc weights of trait-dependent models (+m2 and +m2x versions, Methods) compared with trait-independent models (base model and +x version), averaged across four binary thresholds). Centre line, median; box limits, minimum and maximum; $n = 4$ binary thresholds. The identified biological characteristics of better disperser lineages are indicated by symbols at the tip of the phylogeny, the size of which indicates the maximal difference in dispersal rates between trait states. Clades with life history trade-offs consistent with a fast–slow life history continuum are indicated by * at the end of the life history bars. **d**, Relationship between disperser characteristics in body size and life history analyses. The colours indicate the number of clades in which a given combination of body size–dispersal and life history–dispersal relationships were found. The symbols are the same as in **c**, indicating the shape of the relationships between traits (body size and life history) and dispersal rates.

species were better dispersers in 15% of clades. There were no differences in life history–dispersal shapes among tetrapod classes (Fisher's exact test, $P = 0.84$). These results remained qualitatively similar when we included only clades with differences in dispersal rates between trait states greater than 20%, or when we restricted the analysis to those clades that showed the clearest fast–slow life history continuum

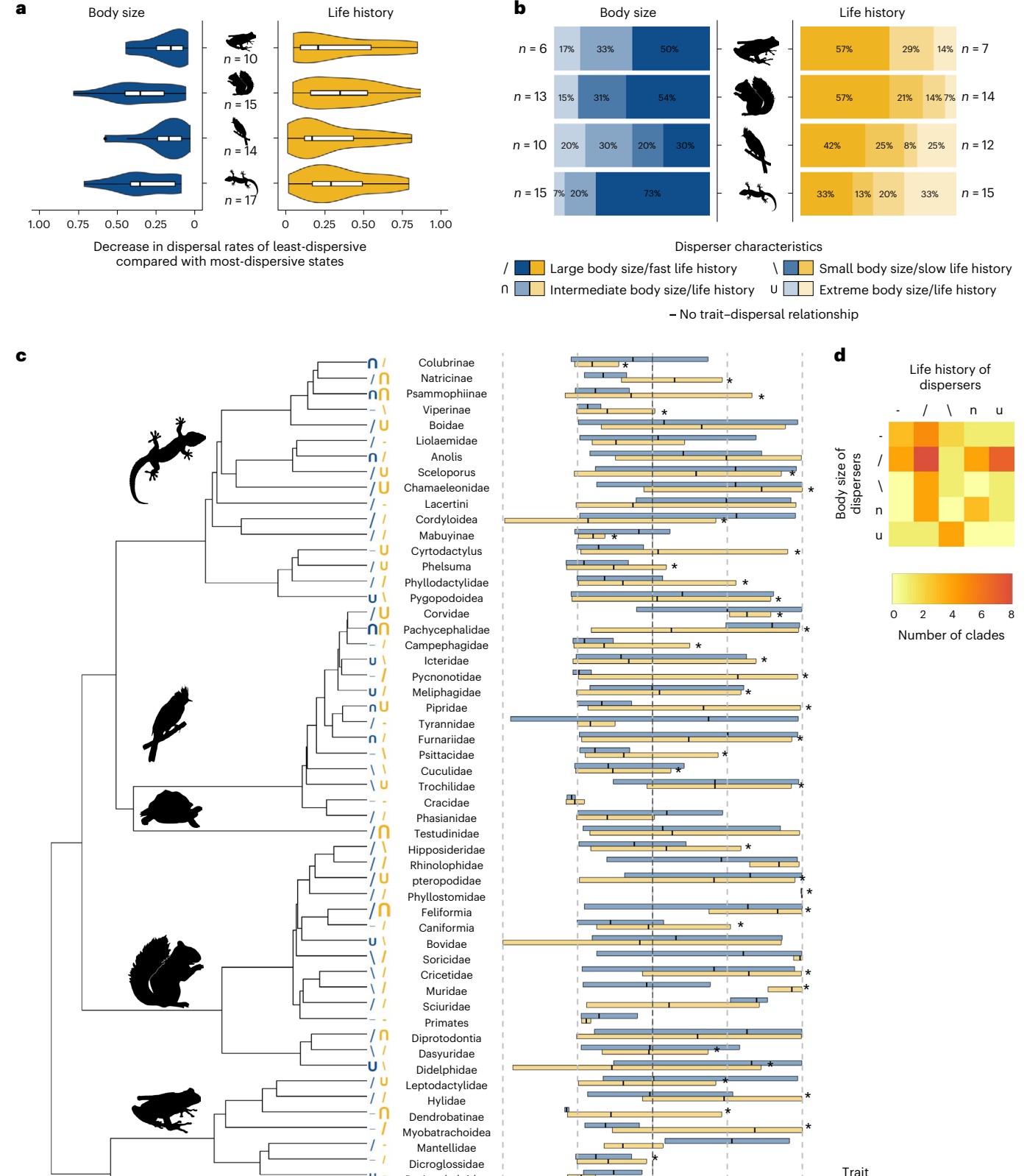

**Table 1 | Ecological and methodological variables tested to understand the magnitude of trait effects on historical dispersal rates across clades**

| Explanatory variables | Hypotheses relating variables to the magnitude of trait effects in historical dispersal rates | Influence on the magnitude of body size effects | Influence on the magnitude of life history effects |
|---|---|---|---|
| **Ecological variables** | | | |
| Variability of body size within clade | Traits with low variability rarely generate large differences in dispersal potential between species[110,111] | | |
| Variability of life history within clade | | | **coef −0.09 (*P*=0.01)** |
| Number of dispersal events per lineage | Proxy for strength of biogeographic barrier. Too few dispersal events decrease statistical power (a problem of small sample size[112]), too many dispersal events indicate weak barriers[113] and thus reduced potential for traits to influence dispersal. This could either lead to a quadratic effect (if both hypotheses are supported) or a linear effect (if only one of them is confirmed) | coef −0.03 (*P*=0.17) | **coef −0.14 (*P*<0.001)** |
| (Number of dispersal events per lineage)² | | coef −0.04 (*P*=0.08) | coef −0.05 (*P*=0.07) |
| Proportion of past oceanic dispersal | Continental dispersal might be less selective than oceanic dispersal | | **coef −0.11 (*P*=0.05)** |
| (Proportion of past oceanic dispersal)² | Oceanic and continental barriers might filter species differently[6,114], which could confound the results when both types of dispersal are combined | | coef 0.07 (*P*=0.08) |
| **Methodological variables** | | | |
| Number of species within clade | Too few species may limit statistical power (a problem of small sample size[112]) | | |
| Trait data coverage | Little available trait data makes it difficult to correctly identify species' life histories, and may obscure trait–dispersal relationships | | |
| Average node resolution | Phylogenetic uncertainty might affect estimations of biogeographic histories | | |
| Crown age | Phylogenetic scale can affect biogeographic estimations; particularly estimations of deep nodes may be biased which might lead to decreased trait effects with increasing clade age[115,116] | | |
| Biogeographic base model | Differences in types of vicariance allowed at a node may introduce variation in dispersal events inferred | | |
| **Full model-adjusted *R*²** | | | |
| | | 5% | 27% |

The hypothesis underlying each variable is described. Effect size (standardized coefficient (coef)) and *P* values are reported if selected in the final models (stepwise variable selection procedure on multivariate regressions, no adjustments for multiple comparisons were made). Trait data coverage was not tested for body size–dispersal relationships since body size was available for all species. Variability of body size within clade was not tested for the effect magnitude of life history, and inversely, variability of life history within clade was not tested for the effect magnitude of body size since we do not assume a causal link between these variables and effects. Variables with a superscript 2 were included to test for quadratic effects. Variables with a *P* value equal or lower than 0.05 are highlighted in bold.

(Extended Data Fig. 3 and Extended Data Table 2). When considering phylogenetic uncertainty, the identification of successful disperser characteristics was robust for some clades, but not for all (for example, Natricinae; Extended Data Fig. 4, Extended Data Table 3, Methods and Supplementary Information Section 1). Future studies should take phylogenetic uncertainty explicitly into account, but such an endeavour is clearly out of the scope of the present study, which is already computationally demanding.

Overall, when analysing the trait–dispersal relationships of the two traits together, large-bodied species were generally more likely to disperse, as well as species with a fast life history strategy, but a dispersal advantage of the two strategies did not necessarily occur simultaneously in the same clades (Fig. 2d). For instance, in some clades, large species with an extreme life history strategy were better dispersers (for example, Chamaeleonidae and Corvidae), and in other clades species with fast life history strategies but small body sizes were better dispersers (for example, Cricetidae and Dasyuridae). Multiple traits may therefore interact in their influence on biogeographic dispersal[33], and this influence may be mediated by the environment and/or biotic interactions[29,30].

Finally, we investigated the potential mechanisms that can explain why the characteristics of the best dispersers vary between clades. We specifically explored the influence of the following ecological and methodological variables and their associated hypotheses:

(1) a clade's average trait value (absolute trait values are more important than relative trait values within clades, for example, the advantage of large size may only be apparent in large-bodied clades), (2) the variability of traits within clades (more complex relationships, for example U- and bell-shaped ones, may only appear in clades with sufficient trait variability), (3) the proportion of observed dispersal events that are oceanic (different traits are linked to different biogeographic barriers), (4) the number of species within a clade (more complex relationships, for example, U- and bell-shaped ones, may only appear in species-rich clades) and (5) trait data coverage (little available trait data makes it difficult to correctly identify species' life histories, and may obscure trait–dispersal relationships). We used a stepwise selection process on multivariate, multinomial regressions to identify the variables that could best explain the different categories of trait–dispersal relationships (positive, negative, bell or U shaped).

Differences in the shape of the body size–dispersal relationships between clades were partly explained by both the average body size and life history of the clades (predictive accuracy, ca. 64%; McFadden *R*² = 0.19; Fig. 3b and Extended Data Table 4). This means that a clade's life history strategy may explain which body sizes are advantageous in dispersal, and that both traits interact to influence dispersal outcomes. A first noteworthy result is that, while large body size was a dispersal advantage in most clades, small size generally was a dispersal advantage in clades with small average body sizes and fast life histories. There

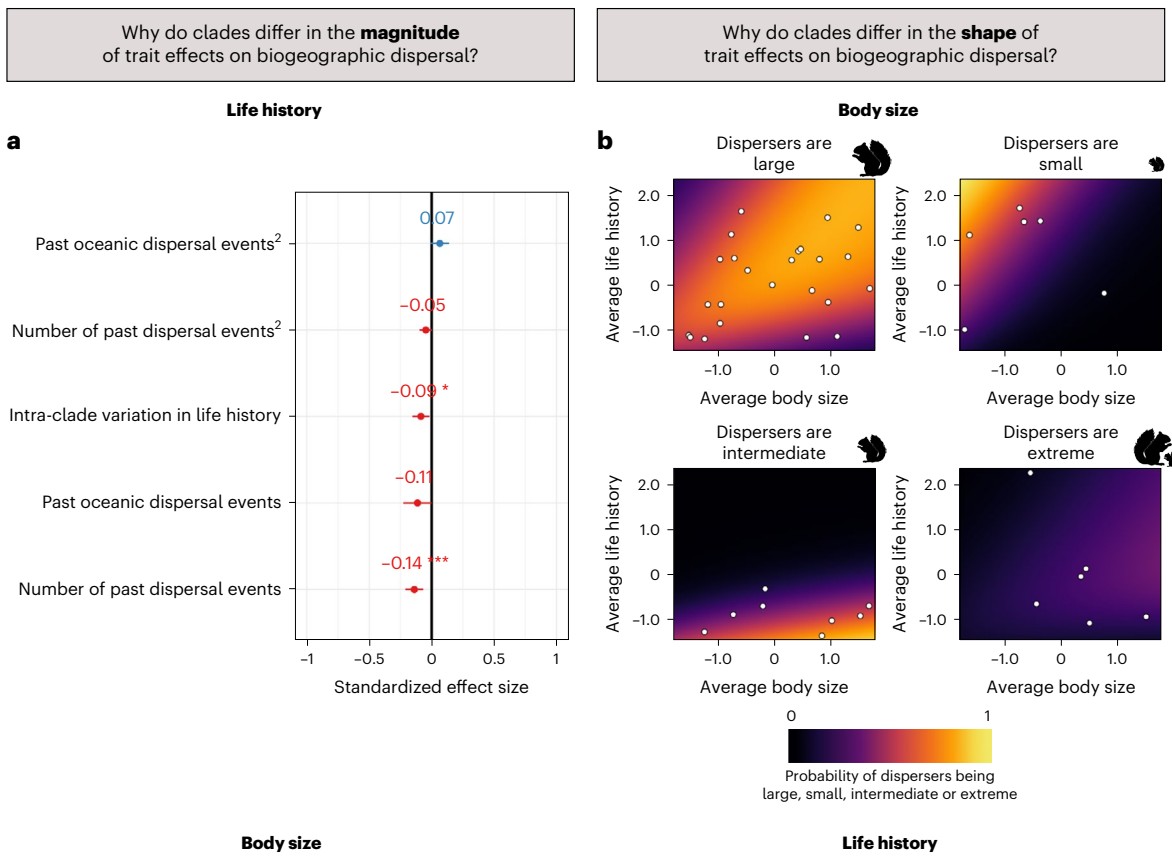

**Fig. 3 | Factors explaining the differences between clades in magnitude and shapes of trait effects. a**, Coefficients (and 95% confidence intervals) of a model regressing the differences in estimated dispersal rates between life history traits across clades against a number of explanatory variables (explained in the text and Table 1). $n = 47$ clades (those where fast–slow consistent trade-offs structured life history). Only variables selected by the model selection procedure are shown (stepwise variable selection procedure on multivariate regressions, no adjustments for multiple comparisons were made), significance is indicated by *$P < 0.05$, ***$P < 0.001$; for exact $P$ values, see Table 1. **b**, Probabilities of different

dispersal shapes depending on a clade's body size and life history relative to other clades of the same tetrapod class. Each point corresponds to one clade where disperser characteristics were as indicated in the title of each panel. The background represents predicted probabilities of specific characteristics being linked to successful dispersal within a clade, where lighter colours correspond to higher probabilities. No variables were significantly related to differences between clades in the magnitude of body size effects on biogeographic dispersal, and no variables were significantly related to differences between clades in shapes of life history–dispersal relationships.

are indeed instances where very small animals have been reported to disperse over long distances (for example, land snails[34] and invertebrates[35]), but this has not been shown for tetrapods before. Small body size could be advantageous in oceanic dispersal because it might increase the probability of several individuals dispersing at once, increasing propagule pressure, which is a strong predictor of colonization success in present-day biological invasions[36]. In clades with relatively fast life histories, small body size might not only be an advantage but rather a constraint: fast-lived species have short lifespans, and if not enough individuals arrive at a far-away location at the same time to reproduce, these individuals might die before a new chance event leads to more individuals arriving. Interestingly, small body size was mainly related to successful dispersal in bird and mammal clades. Only two clades of herpetofauna (10% of notable relationships) showed a U-shaped relationship between body size and dispersal success, and none showed a negative relationship. One possible explanation for these differences between tetrapod classes is that small body sizes lead to higher cooling rates in ectotherms and less thermal stability[37]. This is disadvantageous both for active movement and when passing through cold environments, which one might expect during long-distance dispersal events. Small ectotherms may hence only have a dispersal advantage in specific circumstances, for example, when passing through hot environments where a high cooling rate is desirable.

A second important result is that intermediate sizes were of advantage in clades with overall large body sizes and slow life histories. This indicates that large absolute body sizes do not confer an advantage in biogeographic dispersal if a clade's average life history is very slow. This might be related to lower propagule pressure, since chances might be low of several large individuals dispersing passively at once. In addition, species with slow life histories have generally slow population growth rates, which may lead potential founder populations to fail due to stochastic extinction[13]. When exploring variability in life history–dispersal relationships, none of the tested variables could significantly explain differences between clades (Extended Data Table 4).

In summary, there is substantial variation in trait–dispersal relationships between clades, and in body size–dispersal relationships also between tetrapod classes. Small species, for instance, only had a dispersal advantage in two clades of ectotherms, but in a much larger proportion of mammal and bird clades. We tested several variables that might explain the variation in trait–dispersal relationships, but none of them could explain the variation in life history–dispersal relationships. The shapes of body size–dispersal relationships, on the other hand, were related to the average body size and life history of the entire clade, indicating that both traits may interact in their influence on biogeographic dispersal.

## The future of traits in biogeography

To conclude, we show here for the first time that body size and life history traits are related to, and interact in their influence on, biogeographic dispersal in a large number of tetrapod clades. These traits have thus played a key role in shaping the global biogeography of tetrapods. Traits are part of what determines which lineages have opportunities to colonize and radiate in new biogeographic realms, and which become geographically constrained, making them paramount to understanding historical biogeography and in projecting to the future. The ability of species to overcome large barriers is critical for both climate niche tracking and species' invasions of new biogeographical regions. Both climate change and biological invasions are key challenges of our time because they are key drivers of ecosystem declines and biodiversity shifts[38,39]. Therefore, we urgently need to better understand how species differ in their potential to disperse across biogeographic barriers. We find that traits of successful dispersers vary across clades and we took first steps to find sources of this variation. The magnitude of life history effects on dispersal rates depends on intra-clade trait variability, as well as strength and type (oceanic or continental) of biogeographic barriers. The shape of body size–dispersal relationships, on the other hand, depends on a clade's average body size and life history. These findings are contingent on the methodology we used, which is not without limitations. For instance, it is a known problem that failing to account for geographic distributions of extinction lineages can compromise estimations of biogeographic histories[40]. While there are now methods to model extinct branches[41], these models were impractical here as they remain trait independent and have a very long computation time (computing time of ca. 100 h for a single six-area scenario[41]).

Dispersal capacities are an important determinant of species' vulnerability to global changes; however, additional factors, such as intrinsic sensitivity, need to be considered as well. Adapting our methodology to link species' traits to risk of range contraction could provide important insights in this regard (that is, letting a multiplier modulate local extinction rates, parameter $e$ in BioGeoBEARS models, depending on a lineage's trait state). Considering trait dispersal and trait-range contraction relationships in concert would be a better approximation of species' vulnerability to global changes than assessing dispersal capacities alone.

Despite methodological limitations, our analysis offers exciting perspectives and ideas for future work relating to species' dispersal and colonization. For instance, species' past biogeographic dispersal capacities have been linked to present-day invasion success in plants[42,43], and species' past dispersal capacities might be indicative of species' capacities to track suitable habitat as climate changes. The new knowledge presented here on how traits influence dispersal outcomes could be used to predict and manage biological invasions, improve species distribution modelling predictions and in conservation planning for a rapidly changing world where successful dispersal across major biogeographic barriers is key for species' survival[44].

## Methods

### Taxon selection

We included only clades for which sufficient data was available: for clades that are estimated to contain between 50 and 99 species (according to Bánki et al.[45]) we required more than 60% of data coverage (that is, at least 60% of all species had to have phylogenetic and species distribution data, and trait data for at least one trait; for details, see 'Trait data' section); for clades estimated to contain more than 99 species, we required a coverage of at least 50%.

### Phylogenies

To analyse the role of traits in biogeographic dispersal, we used dated species-level phylogenies of 56 monophyletic clades of amphibians (10 clades), reptiles (17 clades), birds (14 clades) and mammals (15 clades; Supplementary Table 1). We extracted clades where

long-distance dispersal is assumed to have occurred and clades where sufficient data were available from bigger trees (amphibians[46], squamates[47] and mammals[48]). For birds we grafted subclades of the Hackett MCC[49] onto nodes of the backbone of Prum et al.[50], following Cooney et al.[51]. We excluded non-sedentary birds using data from Dufour et al.[52]. Where available, we used clade-specific trees (Supplementary Table 1). In all cases, we excluded taxonomically imputed species from the phylogenies.

For phylogenetic factor analyses at the tetrapod class level (to determine clade positions in the higher-level trait space and intra-clade trait variability, see also 'Explaining the magnitude of trait effects and the shape of trait–dispersal relationships'), we used tetrapod class-level phylogenies: for amphibians the consensus tree[46], and for mammals a node-dated maximum clade credibility tree[48]. To analyse all non-avian reptiles together, we combined the squamate tree[47] with a crocodile[53] and a turtle phylogeny[54]. For birds we used the combined tree as explained above.

### Species distributions

We extracted species distribution data from the International Union for Conservation of Nature (IUCN)[55] for amphibians, mammals and reptiles, and from Birdlife[56] for birds. We only kept records of species where presence was defined as extant, possibly extinct or extinct; origin as native or reintroduced and seasonality as resident or breeding season. For species without a direct match between phylogeny and occurrence data, we used the R package taxize to look up synonyms in the Integrated Taxonomic Information System (ITIS; functions get_tsn it isitis_getrecord, package taxize v0.9.99 (refs. 57,58)). In addition, we used the function synonymMatch (package rangeBuilder v1.5 (ref. 59)), which looks up synonyms in AmphibiaWeb[60], the reptile database[61], the BirdLife Taxonomic Checklist (v8.0 (ref. 62)) and Wilson and Reeder[63]. Where we could not find distribution data for a given species in the IUCN database, we downloaded point occurrence data from the Global Biodiversity Information Facility (https://www.gbif.org/), using the R package rgbif (v.2.2.0 (ref. 64)). We cleaned this data using the package CoordinateCleaner[65]. We then merged the IUCN shapefiles with gbif rasters per clade and reprojected the resulting shapefile into a Behrmann cylindrical equal area projection (for a better comparison between cells close to the poles and cells close to the equator in terms of area). Finally, for each species, we plotted the range map to visually assess it and look for outliers that we removed if present.

### Trait data

For detailed descriptions of data collation, see Supplementary Information Section 3. For amphibians, we combined different trait databases[66–69] and extracted snout–vent length (SVL), egg size, clutch size, clutches per year, age at sexual maturity (SM) and longevity (LG). For non-avian reptiles, we combined data[66,67,70–74] and extracted SVL, hatchling body mass, clutch size, clutches per year, SM and LG. For mammals, we combined trait databases[73,75–84] and extracted body mass (BM), neonate body mass, litter size, litters per year, LG, SM, gestation time and weaning age. Finally, for birds we combined data[52,73,77,80,81,85–103] and extracted BM, egg mass, litter size, litters per year, LG, SM, gestation time and weaning age.

### Matching phylogenetic and trait data

The phylogenies and combined trait databases did not follow the same taxonomies. We therefore looked up synonyms for all species without a direct match in the trait databases. To do so, we again used the ITIS (functions get_tsn and itis_getrecord, package taxize v0.9.99 (refs. 57,58)) and the package rangeBuilder (v1.5 (ref. 59); function synonymMatch), which looks up synonyms in AmphibiaWeb[60], the reptile database[61], the BirdLife Taxonomic Checklist (v8.0 (ref. 62)) and Wilson and Reeder[63]. In addition, we looked up by hand species without a match after this process. Where several matches in the

trait databases for a species with phylogenetic data were found, we averaged the trait values.

## Bioregionalization and paleoreconstruction

We used biogeographic models implemented in BioGeoBEARS (DEC, DIVALIKE and BAYAREALIKE[94]) to estimate clades' biogeographic histories. These models infer past range evolution, that is, dispersal, vicariance and range contractions, over discrete areas that need to be defined a priori. For every clade, we thus identified relevant biogeographic barriers by calculating phylogenetic beta-diversity[95,96] between raster cells (using an equal area projection, Behrmann projection with standard parallels at 30° to make raster cells between higher and lower latitudes comparable). To do this, we used the R package betapart v1.5.6 (ref. [97]). We weighted the phylogenetic beta-diversity matrix with a geographical distance matrix (great-circle distances on latitude/longitude coordinates) and applied an unweighted pair group method with arithmetic mean following the methodology in Weil et al.[11]. We reconstructed movement of biogeographic regions using Gplates[98] and a global plate and rotation model[99]. On the basis of those reconstructions, we implemented a time-stratified analysis if necessary to reflect plate tectonic movements and island uplifts in the biogeographic estimations. In addition, we implemented root constraints where strong indications for the root distribution were available (Supplementary Table 1). All biogeographic models implemented in BioGeoBEARS estimate rates of range contraction (extinction parameter $e$) and range expansion (dispersal parameter $d$). In very basic models, the dispersal rate $d$ is the same for all lineages and between all regions. However, this is unrealistic, as barriers separating regions may influence dispersal rates. Therefore, $d$ can be modified through manual dispersal multiplier matrices (MDMMs), which are matrices applying multipliers to dispersal rates between pairs of regions (one matrix per time slice in time-stratified analyses). As dispersal across oceanic barriers may be less likely than dispersal across continental barriers, we implemented three different MDMMs, adapted from Weil et al.[11]. We defined these MDMMs on the basis of the assumption that continental dispersal is more likely than trans-oceanic dispersal. Using biogeographic models implemented in BioGeoBEARS, we used model comparison based on AICc weight to identify the best MDMM for each clade (choosing between (1) an equal weight MDMM with no difference between continental and oceanic dispersal probabilities, (2) a 0.5 version where continental dispersal was set to 0.5 and oceanic dispersal to 0.125 and (3) a 0.1 version where the probability of oceanic dispersal was set to 0.05, that is ten times less likely than continental dispersal) (for which model and MDMM was chosen for each clade, see Supplementary Table 1). Dispersal rates may differ not only between regions but also between lineages, for example, depending on their traits. This is also implemented in BioGeoBEARS (for details, see 'Trait–dispersal relationships').

## PFA

To detect latent variables that structure life histories across taxonomic groups, we used a PFA[22] that allows for missing data (for more details, see Hassler et al.[100] and Weil et al.[11]). This PFA is implemented in the Julia package PhylogeneticFactorAnalysis.jl v0.1.4 (ref. [23]), which relies on a development version of BEAST[101] to be released with BEAST v1.10.5. Body size can influence life history traits through allometric constraints while evolving under different selection pressures. We therefore structured the PFA so that body size (as captured by SVL for amphibians and non-avian reptiles, and BM for birds and mammals) loaded only onto the first factor while all other traits loaded onto all factors, following Weil et al.[11]. Hence, life history trait variation associated with body size was forced onto the first factor and the second factor we extracted captured size-independent patterns of life history covariation. We analysed all clades separately, as trade-offs structuring species' life histories can vary between clades and biogeographic models were conducted at the clade level. We then repeated the PFA analyses separately for all amphibians, mammals, birds and reptiles together to determine

position and variation of individual clades in the higher-level trait space (data coverage per clade; Supplementary Table 2).

## Trait–dispersal relationships

As in Weil et al.[11], we first determined the best base model (DEC, DIVALIKE or BAYAREALIKE) and manual dispersal multiplier matrix (MDMM null, MDMM 0.5 or MDMM 0.1) for each clade. We then ran trait-dependent and trait-and-distance-dependent models only on the best base model applying the most supported MDMM (for which combination was chosen for which clade, see Supplementary Table 1). In trait-dependent models, the dispersal probability of a lineage is multiplied by a parameter $m$, depending on which trait state the lineage is in. Due to computational constraints, most studies of trait-dependent biogeography currently only include binary traits. In practice, the multiplier of trait state 1, parameter $m_1$, is fixed to 1 and the multiplier of trait state 2, parameter $m_2$, is estimated by the model. If $m_2$ is greater than 1, this indicates that trait state 2 is positively related to dispersal, and inversely, if $m_2$ is smaller than 1, it indicates that trait state 1 is positively related to dispersal. Since the inclusion of binary traits can only offer limited information, we tested four different binarizations: for each trait, we tested the first 50% of species against the rest (that is, using a median split on the factors of the PFA), the first 25% against the rest, the last 25% against the rest and the extreme 50% (first and last 25%, taken together against the rest). We then calculated weighted averages of the dispersal multipliers for the different intervals on the two factors of the PFA to obtain an estimate of the relationship between trait (that is, factor) and dispersal probability for each clade.

To quantify the magnitude of the effect of traits on dispersal rates, we calculated the maximal difference in dispersal rates between trait states per clade. We then determined the exact shape of the relationship for those clades in which the maximal difference in dispersal rates between trait states exceeded 10%, separately for body size and life history strategy. We distinguished between positive (large/fast-lived species were better dispersers than small/slow-lived species), negative (small/slow-lived species were better dispersers than large/fast-lived species), bell-shaped (species with intermediate traits were better dispersers than species with extreme traits) and U-shaped relationships (species with extreme traits were better dispersers than species with intermediate traits).

To assess the effect of phylogenetic uncertainty on the classification of trait–dispersal relationships, we selected ten trees for one reptile, one amphibian and one mammal clade and repeated phylogenetic factor analyses, as well as trait-independent and trait-dependent biogeographic models (Extended Data Fig. 4 and Table 4) for all individual trees, using the best base model and MDMM, as identified in the main analysis. For one bird clade, we repeated all analyses on a tree where we used the backbone of Prum et al.[50] that included the fossil vegavis.

The computation time of all biogeographic analyses combined amounted to ca. 262,800 h × cores of calculations on a high-performance computing cluster (GRICAD infrastructure, https://gricad.univ-grenoble-alpes.fr), corresponding to emissions of ca. 1.2 t of $CO_2$eq.

## The effect of traits in biogeographic estimations

To assess how the inclusion of traits impacted biogeographic estimations, we analysed the following three metrics: (1) the average resolution of ancestral range estimates, (2) the average number of dispersal events in a clade's biogeographic history and (3) the proportion of nodes that were estimated differently between trait-dependent and trait-independent models (and between distance-dependent and distance-independent models, as a comparison).

(i) The average resolution of ancestral range estimates for each clade was calculated in two steps. First, we calculated AICc-weighted averages for ancestral range estimations at

each node of the phylogeny, separately for trait-dependent and trait-independent models (+m2 and +m2x models versus base model and +x model). We repeated this for each trait (both body size and life history) and binarization split (four splits: median split, first 25%, last 25% and extreme 25%). To have a point of comparison, we also calculated AICc-weighted averages for distance-dependent and distance-independent models (+x and +m2x versus base model and +m2 models). Second, we calculated mean resolutions of ancestral range estimations, that is, the proportion of most likely range per node divided by the number of nodes in the phylogeny, excluding tips. The results were averaged across binarization splits per trait, which led to eight values per clade in total: two traits (body size and life history) × four model groups (trait dependent, trait independent, distance dependent and distance independent).

(ii) To calculate the total number of dispersal events in the biogeographic history of a clade, we created 100 biogeographic stochastic maps[102] for all models (base model, base model+x, base_model+m2 and base_model+m2x) and all binarization splits (median split, first 25%, last 25% and extreme 25%, for both body size and life history). We counted the total number of dispersal events for each of the 100 maps we created per model, trait and binarization split (1,800 maps in total), and divided it by the number of species in the clade to be able to compare values between clades. We then calculated AICc-weighted averages for trait-dependent and trait-independent, and distance-dependent and distance-independent models, and averaged across binarization splits per trait (which again led to eight values per clade, as above).

(iii) Finally, we calculated the proportion of nodes that were estimated differently between the different groups of models (trait-dependent versus trait-independent and distance-dependent versus distance-independent models). To do so, we first determined the most likely ancestral range for each node in each model group for each binarization split. We then calculated differences between trait-dependent and trait-independent models, as well as distance-dependent and distance-independent models, and averaged across binarization splits per trait. This led to four values per clade (two traits (body size and life history) × two comparisons (trait dependent versus trait independent and distance dependent versus distance independent)).

We used mixed-effects models (R packages lme4 v1.1.30 (ref. 103) and car v3.1.0 (ref. 104)) to assess how resolution and number of dispersal events in a clade's biogeographic history varied between the different model groups (trait-dependent, trait-independent, distance-dependent and distance-independent estimations), using clade identity as a random effect. Where these models were significant, we followed with contrasts among estimated marginal means (contrasting trait-dependent and trait-independent models, distance-dependent and distance-independent models, distance-dependent and trait-dependent models, and distance-independent and trait-independent models, using a Šidák correction for multiple comparisons, in R package emmeans v1.8.2 (ref. 105)). We checked the normality of residuals visually with histograms, and homoscedasticity by plotting residuals against fitted values.

### Explaining differences between clades

We first used linear regressions to test whether the differences between clades in the maximal difference in dispersal rates between trait states could be explained by (1) the total average number of dispersal events within a clade' biogeographic history, (2) the percentage of oceanic dispersal events in a clade's biogeographic history, (3) the intra-clade variability in body size or life history, (4) a clade's species richness, (5) data coverage, (6) average node resolution across the phylogeny,

(7) clades' crown age and (8) the biogeographic base model (DEC, BAYAREA or DIVA). To understand why trait–dispersal relationships vary across clades, we performed multinomial logistic regressions to test if the different categories of relationships (positive, negative, bell shaped and U shaped) could be explained by (1) a clade's average body size or life history strategy, (2) intra-clade variability in body size or life history, (3) the percentage of oceanic dispersal events in a clade's biogeographic history, (4) a clade's species richness and (5) data coverage (hypotheses relating the individual variables to the magnitude of trait effects or trait–dispersal relationships can be found in the main text and in Table 1). For multinomial regressions, we used the R packages nnet v7.3.17 (ref. 106) and mlogit v1.1.1 (ref. 107). In both cases, we used a stepwise selection process (in both directions, R package MASS v7.3.55 (ref. 106)) to identify those variables that were significantly related to the variables of interest. In the first case, where we tested for differences between clades in the maximal difference in dispersal rates between trait states, we used a two-step approach as there were too many predictor variables for the low number of observations (56 clades). We first tested the methodological variables (variables 4–8) and retained only those which were selected in the step-selection process for the following model, where we included the ecological variables (variables 1–3). There was no phylogenetic signal in the residuals of any of the models (function 'phylosig', R package phytools v1.2-0 (ref. 108)); therefore, there was no need to account for phylogenetic relationships between clades. To determine the average position of all clades in the higher-level trait space and intra-clade trait variability, we conducted tetrapod class-level phylogenetic factor analyses for amphibians, mammals, reptiles and birds. We scaled clades' median positions in those higher tetrapod class trait spaces between −1 and 1, with 0 coinciding with the median position of the entire tetrapod class. We expressed intra-clade trait variability as a percentage of variability in the entire tetrapod class. To calculate the total number of dispersal events in the biogeographic history of a clade, as well as the percentage of oceanic dispersal, we created 100 biogeographic stochastic maps[103] for all models (base model, base model+x, base_model+m2 and base_model+m2x) and all binarization splits (median split, first 25%, last 25% and extreme 25%). We counted the number of oceanic dispersal events and the total number of dispersal events for each of the 100 maps we created per model and binarization split, averaged them across the 100 maps and then calculated an AICc-weighted average per binarization split. Finally, we averaged the number of oceanic dispersal events and the total number of dispersal events across binarization splits, and expressed the number of oceanic dispersal events as a percentage of the total number of dispersal events.

Since the distributions of the variables total average number of dispersal events, the percentage of oceanic dispersal events and clades' species richness were skewed, we log-transformed these variables before conducting the regressions. All variables were scaled before the regressions to compare their influence in a single model. Predictor variables that were included in a single model were generally weakly correlated ($-0.35 <$ Pearson correlation $< 0.35$), with some exceptions: crown age was moderately correlated with intra-clade trait variability (Pearson correlation 0.44 and 0.54 for life history and body size, respectively); the average number of dispersal events was moderately correlated with the percentage of oceanic dispersal (Pearson correlation $-0.41$), as was average node resolution (Pearson correlation 0.46). The average number of dispersal events was strongly correlated with average node resolution (Pearson correlation $>0.9$), indicating that a lot of movement in biogeographic histories leads to more uncertain estimations. The R package 'ggtree' v3.6.2 (ref. 109) was used to prepare Figs. 1 and 2.

### Reporting summary

Further information on research design is available in the Nature Portfolio Reporting Summary linked to this article.

## Data availability

Data for analyses of historical biogeography (trait dependent and otherwise), as well as data related to the analysis of differences in the magnitude of trait effects and differences in trait–dispersal patterns are archived on Figshare at https://doi.org/10.6084/m9.figshare.21897003. Further data, for preliminary and intermediate analyses, will be made available by the authors upon reasonable request.

## Code availability

R scripts for analyses of historical biogeography (trait dependent and otherwise), as well as code related to the analysis of differences in the magnitude of trait effects and differences in trait–dispersal patterns are archived on Figshare at https://doi.org/10.6084/m9.figshare.21897003. Further code for preliminary and intermediate analyses will be made available by the authors upon reasonable request.

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

## Acknowledgements

We thank J. Renaud for providing part of the species distribution data, and B. Bzeznik and M. Guéguén for assistance with the cluster. M. González-Suárez, S. Diamond and F. Boucher provided valuable insight during discussions. All BioGeoBEARS analyses were performed using the GRICAD infrastructure (https://gricad.univ-grenoble-alpes.fr), which is supported by Grenoble research communities. We thank Swansea University for covering the open access fees. This work is the result of a collaboration between Université Grenoble Alpes and Swansea University, supported by Initiative d'excellence (IDEX) International Strategic Partnership and Swansea University Strategic Partner Research (SUSPR) scholarships.

## Author contributions

L.G. and W.L.A. conceived the project. L.G., W.L.A. and S.-S.W. conceptualized the study. S.-S.W., with the help of W.L.A. and M.P.J.N., collected the data. S.-S.W., with the help of M.P.J.N., performed bioregionalizations. S.-S.W. performed all further analyses and wrote the original draft of the manuscript. L.G., W.L.A., M.P.J.N., S.L. and L.B. reviewed and edited the manuscript. L.G. and W.L.A. acquired financial support for the project.

## Competing interests

The authors declare no competing interests.

## Additional information

**Extended data** is available for this paper at https://doi.org/10.1038/s41559-023-02150-5.

**Correspondence and requests for materials** should be addressed to Sarah-Sophie Weil.

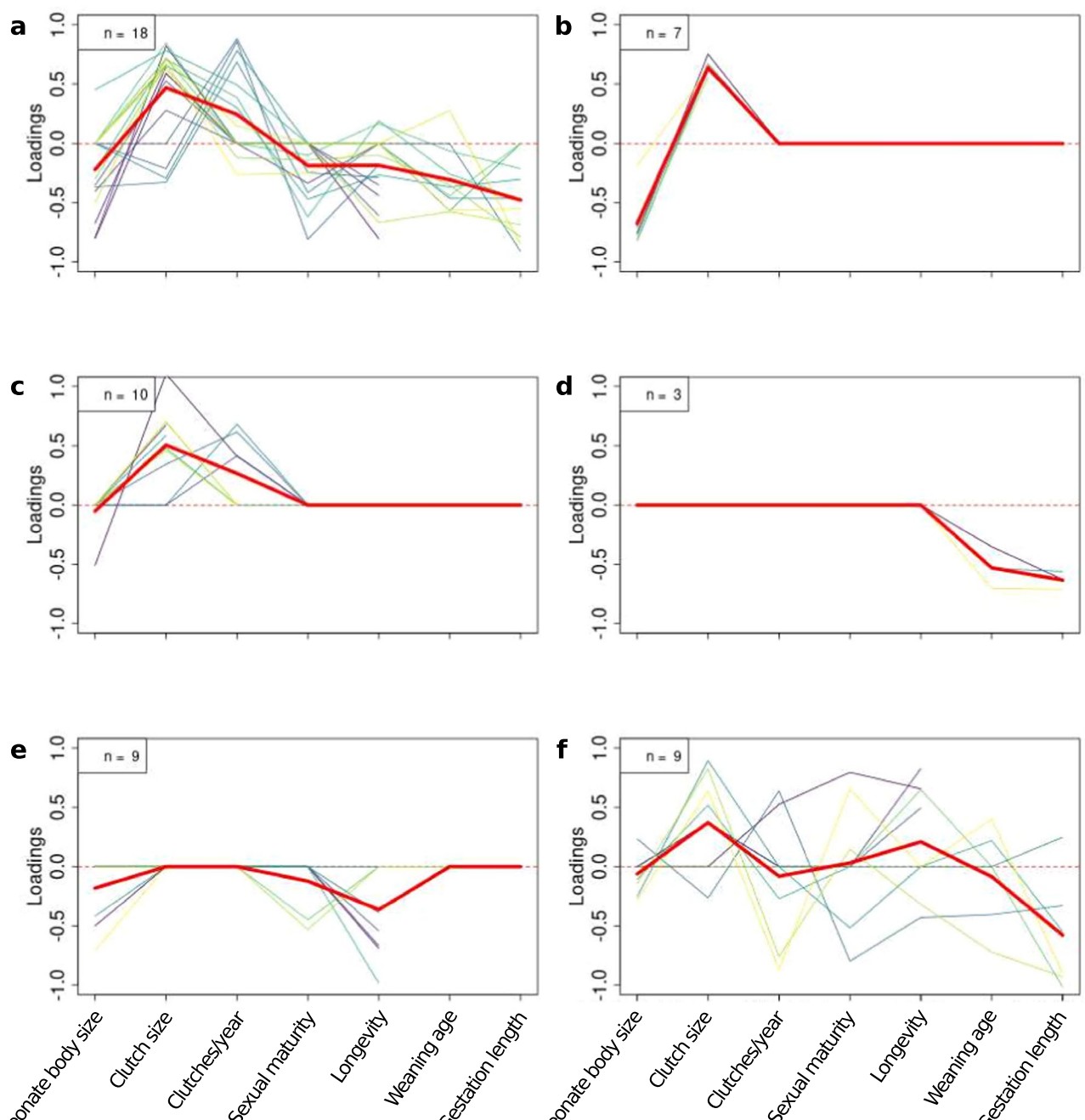

**Extended Data Fig. 1 | Summary of variable loadings on the second factor of phylogenetic factor analyses, grouped by interpretation of main life history trade-offs.** One thin line corresponds to one clade; the thick red line is the average of all clades in a panel. a) Clades in which we found a clear fast-slow life history spectrum (where fast species reproduced quickly and were short-lived and slow species had opposing traits; 18/56 clades=32%). b), c), d) Clades in which other fast-slow consistent trade-offs structured life history (20 clades=36%). e) Clades in which only one trait loaded onto the second factor (9 clades=16%). f) Clades in which trait covariation did not suggest any trade-off between survival and reproduction consistent with the fast-slow life history spectrum (9 clades=16%).

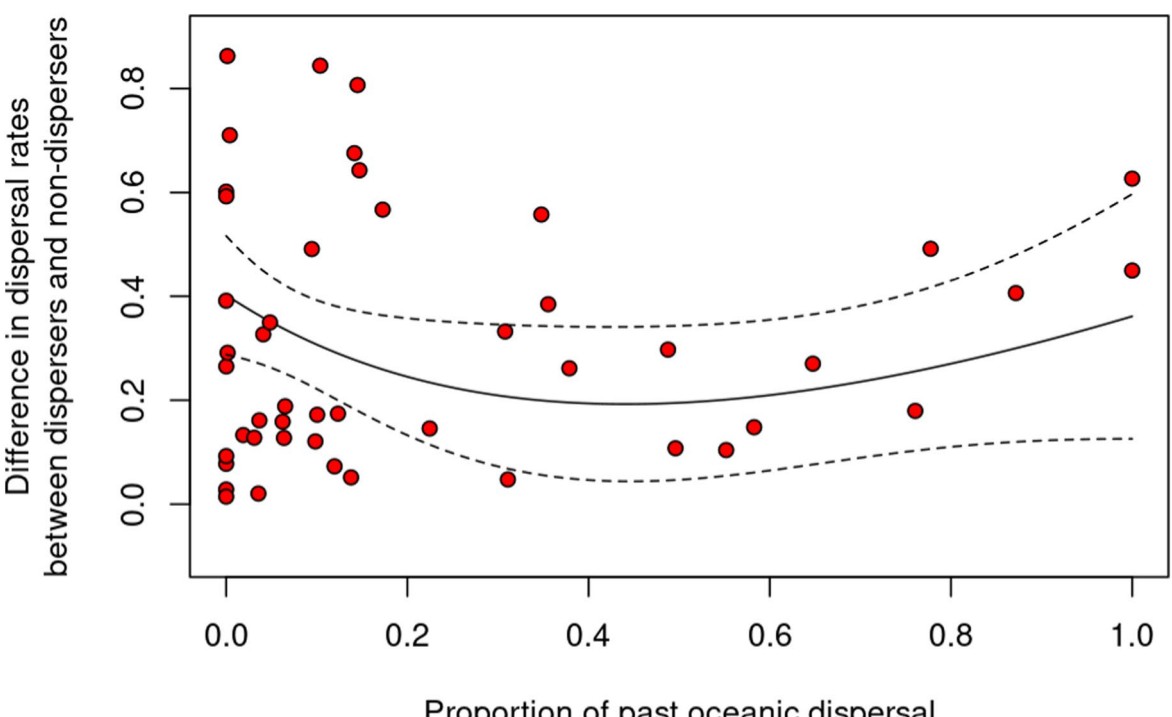

**Extended Data Fig. 2 | Relationship between proportion of past oceanic dispersal and magnitude of life history effects on dispersal rates.** Each point corresponds to a clade; the line was fitted based on the selected model explaining differences between clades in dispersal rate differences between most-dispersive and least-dispersive lineages. Dotted lines correspond to confidence intervals.

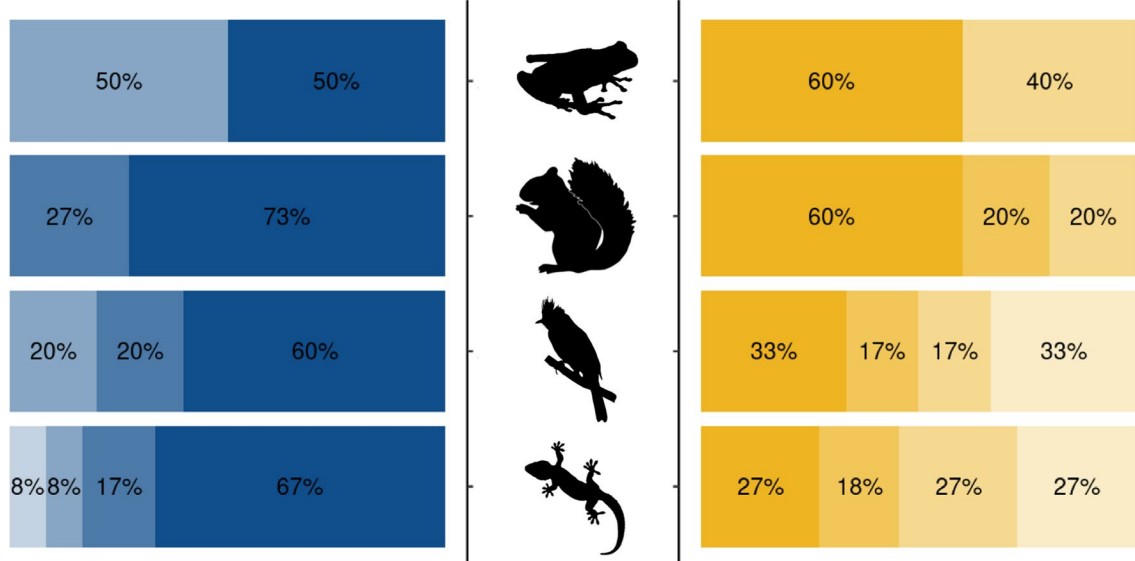

## Disperser characteristics

- large/fast life history
- small/slow life history
- intermediate traits
- extreme traits

**Extended Data Fig. 3 | Distribution of trait-dispersal relationships by tetrapod class.** Only clades where the maximal difference in dispersal rates between most-dispersive and least-dispersive trait states was greater than 20% are included (4 amphibian clades, 5 bird clades, 11 mammal clades, 12 reptile clades for body size-dispersal relationships; 5 amphibian clades, 6 bird clades, 10 mammal clades, 11 reptile clades for life history-dispersal relationships (only including clades where a fast-slow life history spectrum was identified)).

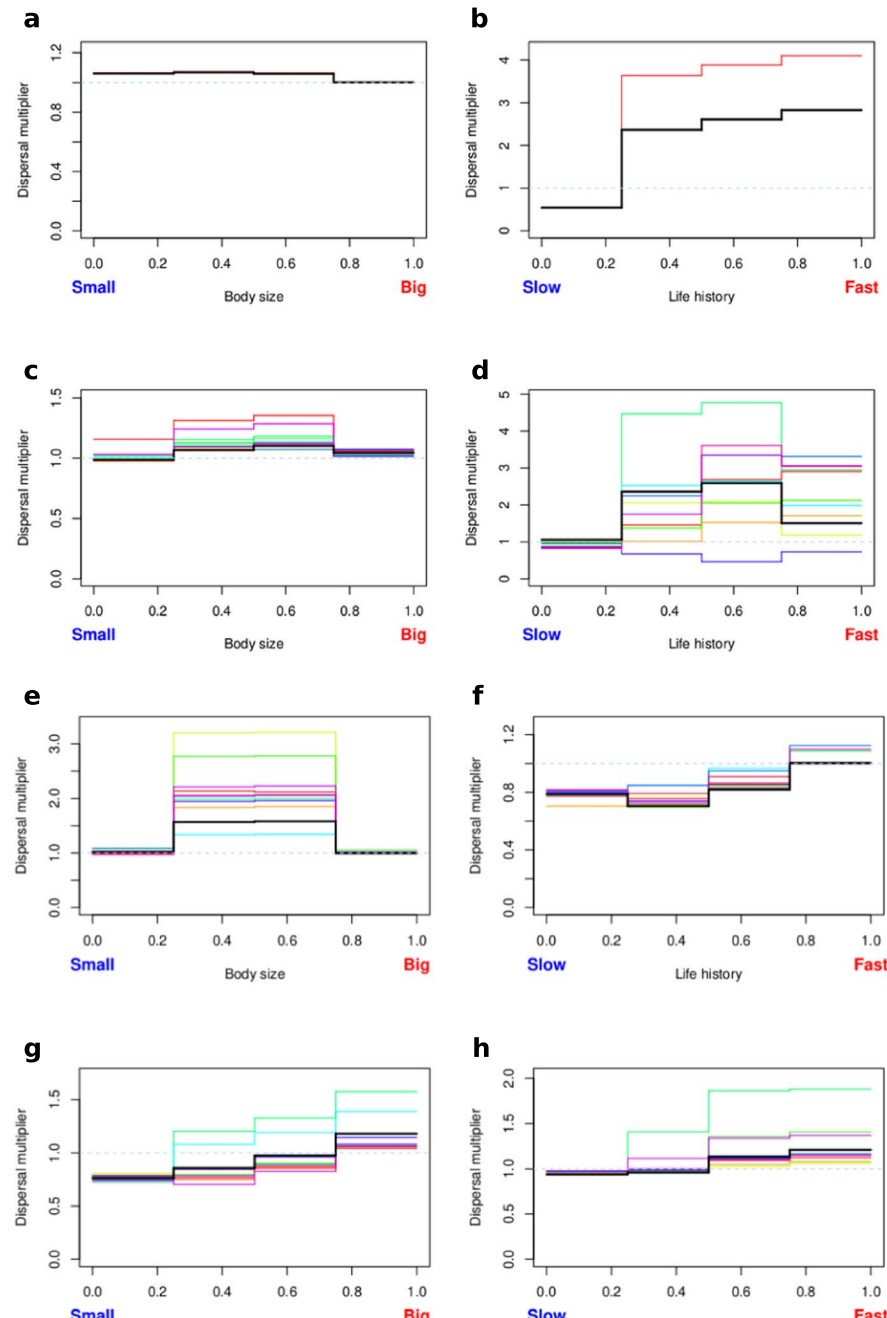

**Extended Data Fig. 4 | Effect of phylogenetic uncertainty on trait-dispersal relationships.** In black: the trait-dispersal relationship we identified with the maximum consensus tree; in different colours: the trait-dispersal relationships when different phylogenetic trees from the posterior were used (methodology to choose those trees as in Weil et al.[11]). a) and b) Pycnonotidae, c) and d) Natricinae, e) and f) Hynobiidae, g) and h) Sciuridae.

**Extended Data Table 1 | Model selection results for the magnitude of trait effects**

| | Variables | Body size (56 clades) | Life history 1 (47 clades) | Life history 2 (38 clades) |
|---|---|---|---|---|
| **Ecological variables** | Variability of body size within clade | | n.t. | n.t. |
| | Variability of life history within clade | n.t. | coef = -0.09 (pval = 0.01) | coef = -0.11 (pval = 0.009) |
| | Number of dispersal events per lineage | coef = -0.03 (pval = 0.17) | coef = -0.14 (pval < 0.001) | coef = -0.12 (pval = 0.002) |
| | Number of dispersal events per lineage$^2$ | coef = -0.04 (pval = 0.08) | coef = -0.05 (pval = 0.07) | coef = -0.04 (pval = 0.18) |
| | Proportion of past oceanic dispersal | | coef = -0.11 (pval = 0.05) | |
| | Proportion of past oceanic dispersal$^2$ | | coef = 0.07 (pval = 0.08) | |
| **Methodological variables** | Number of species | | | |
| | Trait data coverage | n.t. | | |
| | Average node resolution | | [coef = 0.07 (pval = 0.04)] | [coef = 0.08 (pval = 0.05)] |
| | Crown age | | | |
| | Biogeographic base model | | | |
| | Adjusted R$^2$ | 5% (pval 0.11) | 27% (pval 0.003) | 26% (pval 0.004) |

We conducted multiple linear regressions to explain the effect of several variables (in rows) on the differences between clades in the maximal differences in dispersal rates between trait states (in columns). Due to the large number of variables compared to observations, we took a two-step approach: first, we included only methodological variables in the regressions, then we included ecological variables and those methodological variables that were selected in the first step (no adjustments for multiple comparisons were made). For variables that were selected in the stepwise model selection process we give coefficients and p-values of variables. Variables in square brackets were selected in the regression with only methodological variables, but not in the final regression. Variables that were not tested for a given dataset because we did not hypothesize them to causally influence the magnitude of trait effects are indicated by n.t. Life history 1 is the dataset presented in the main manuscript, including 47 clades in which we identified a fast-slow life history continuum (Extended Data Fig. 2); in Life history 2 we excluded clades where only one variable loaded onto the second factor of the phylogenetic factor analysis.

**Extended Data Table 2 | Sensitivity of trait-dispersal relationships to different inclusion thresholds and life history datasets**

| Dataset | Identified relationships | | | | | Number of clades |
|---|---|---|---|---|---|---|
| | NONE | POS | NEG | BELL | U | |
| Body size 10% | 12 (21%) | 24 (43%) | 6 (11%) | 8 (14%) | 6 (11%) | 56 |
| Life history 1 10% | 8 (17%) | 17 (36%) | 7 (15%) | 6 (13%) | 9 (19%) | 47 |
| Life history 2 10% | 7 (18%) | 16 (42%) | 5 (13%) | 4 (11%) | 6 (16%) | 38 |
| Body size 20% | 24 (43%) | 21 (38%) | 6 (11%) | 4 (7%) | 1 (2%) | 56 |
| Life history 1 20% | 22 (47%) | 10 (21%) | 5 (11%) | 6 (13%) | 4 (9%) | 47 |
| Life history 2 20% | 18 (47%) | 10 (26%) | 4 (11%) | 4 (11%) | 2 (5%) | 38 |

10% (20%) indicates the threshold of maximum difference in dispersal rates between trait states below which we decided not to identify the shape of a relationship. Life history 1 is the dataset presented in the main manuscript, including 47 clades in which we identified a fast-slow life history continuum (Extended Data Fig. 2); in Life history 2 we excluded clades where only one variable loaded onto the second factor of the phylogenetic factor analysis.

**Extended Data Table 3 | Assessing phylogenetic uncertainty in trait-dispersal relationships**

| Clade | Trait | Identified relationships | | | | | Correctly identified |
|---|---|---|---|---|---|---|---|
| | | NONE | POS | NEG | BELL | U | |
| Hynobiidae | Body | 0 | 0 | 0 | 10 | 0 | 100% |
| Hynobiidae | LH | 0 | 10 | 0 | 0 | 0 | 100% |
| Sciuridae | Body | 0 | 10 | 0 | 0 | 0 | 100% |
| Sciuridae | LH | 2 | 8 | 0 | 0 | 0 | 80% |
| Pycnonotidae | Body | 1 | 0 | 0 | 0 | 0 | 100% |
| Pycnonotidae | LH | 0 | 1 | 0 | 0 | 0 | 100% |
| Natricinae | Body | 2 | 5 | 0 | 3 | 0 | 50% |
| Natricinae | LH | 0 | 5 | 0 | 4 | 1 | 40% |

None indicates that the differences in dispersal rates between most-dispersive and least-dispersive lineages was less than 10%. POS: large/fast-lived species were better dispersers than small/slow-lived species; NEG: small/slow-lived species were better dispersers than large/fast-lived species; BELL: species with intermediate traits were better dispersers than species with extreme traits; U: species with extreme traits were better dispersers than species with intermediate traits.

**Extended Data Table 4 | Model selection results for the shape of trait-dispersal relationships**

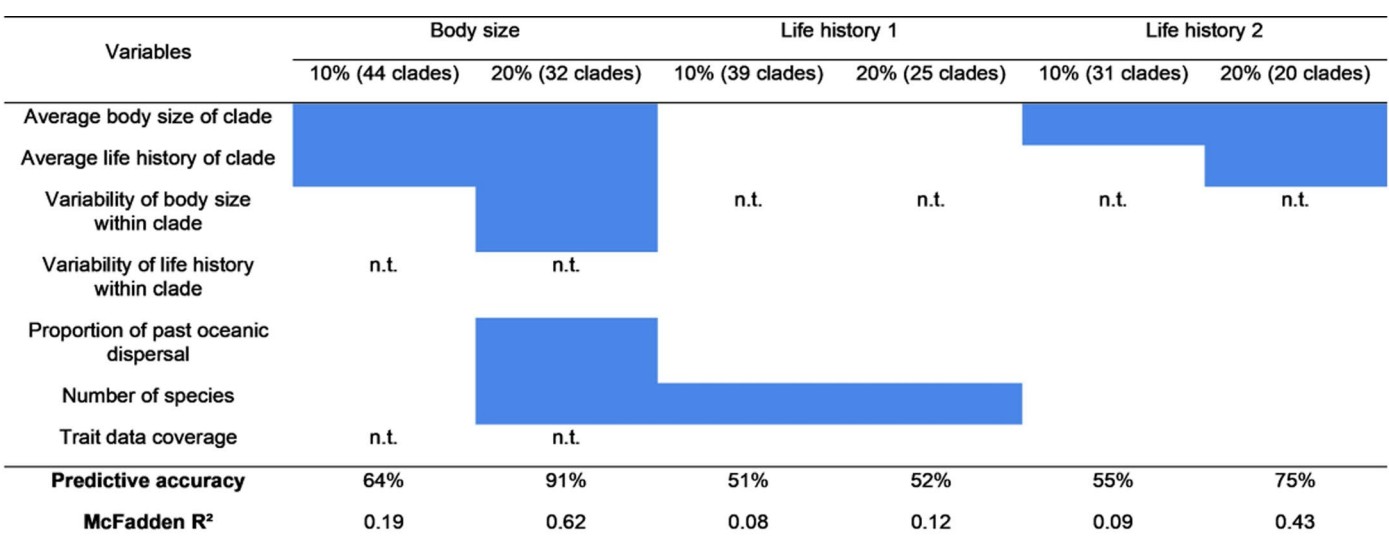

| Variables | Body size | | Life history 1 | | Life history 2 | |
|---|---|---|---|---|---|---|
| | 10% (44 clades) | 20% (32 clades) | 10% (39 clades) | 20% (25 clades) | 10% (31 clades) | 20% (20 clades) |
| Average body size of clade | | | | | | |
| Average life history of clade | | | | | | |
| Variability of body size within clade | | | n.t. | n.t. | n.t. | n.t. |
| Variability of life history within clade | n.t. | n.t. | | | | |
| Proportion of past oceanic dispersal | | | | | | |
| Number of species | | | | | | |
| Trait data coverage | n.t. | n.t. | | | | |
| **Predictive accuracy** | 64% | 91% | 51% | 52% | 55% | 75% |
| **McFadden R²** | 0.19 | 0.62 | 0.08 | 0.12 | 0.09 | 0.43 |

We used multinomial regressions to assess the effect of several variables (in rows) on the trait-dispersal relationships in different datasets (in columns). Variables that were selected are highlighted in blue; variables that were not tested for a given dataset because we did not hypothesize them to causally influence the magnitude of trait effects are indicated by n.t. Life history 1 is the dataset presented in the main manuscript, including 47 clades in which we identified a fast-slow life history continuum (Extended Data Fig. 2); in Life history 2 we excluded clades where only one variable loaded onto the second factor of the phylogenetic factor analysis. 10% (20%) indicates the threshold of maximum difference in dispersal rates between trait states below which we decided not to identify the shape of a relationship.

# Reporting Summary

## Statistics

For all statistical analyses, confirm that the following items are present in the figure legend, table legend, main text, or Methods section.

| n/a | Confirmed | |
|---|---|---|
| ☐ | ☒ | The exact sample size (*n*) for each experimental group/condition, given as a discrete number and unit of measurement |
| ☐ | ☒ | A statement on whether measurements were taken from distinct samples or whether the same sample was measured repeatedly |
| ☐ | ☒ | The statistical test(s) used AND whether they are one- or two-sided *Only common tests should be described solely by name; describe more complex techniques in the Methods section.* |
| ☐ | ☒ | A description of all covariates tested |
| ☐ | ☒ | A description of any assumptions or corrections, such as tests of normality and adjustment for multiple comparisons |
| ☐ | ☒ | A full description of the statistical parameters including central tendency (e.g. means) or other basic estimates (e.g. regression coefficient) AND variation (e.g. standard deviation) or associated estimates of uncertainty (e.g. confidence intervals) |
| ☐ | ☒ | For null hypothesis testing, the test statistic (e.g. $F$, $t$, $r$) with confidence intervals, effect sizes, degrees of freedom and $P$ value noted *Give P values as exact values whenever suitable.* |
| ☒ | ☐ | For Bayesian analysis, information on the choice of priors and Markov chain Monte Carlo settings |
| ☒ | ☐ | For hierarchical and complex designs, identification of the appropriate level for tests and full reporting of outcomes |
| ☒ | ☐ | Estimates of effect sizes (e.g. Cohen's *d*, Pearson's *r*), indicating how they were calculated |

*Our web collection on statistics for biologists contains articles on many of the points above.*

## Software and code

Policy information about availability of computer code

| Data collection | No software was used to collect data. R v3.6.3 (<www.r-project.org>) was used to compile different databases |
|---|---|
| Data analysis | R v3.6.2 (<www.r-project.org>) was used for biogeographic analyses on the high-performance computation cluster, R v3.6.3 (<www.r-project.org>) was used for all further analyses. Data and code to support our results are deposited on Figshare. A doi is included in the data and code availability statements in the main manuscript file. |

For manuscripts utilizing custom algorithms or software that are central to the research but not yet described in published literature, software must be made available to editors and reviewers. We strongly encourage code deposition in a community repository (e.g. GitHub). See the Nature Portfolio guidelines for submitting code & software for further information.

## Data

Policy information about availability of data

All manuscripts must include a data availability statement. This statement should provide the following information, where applicable:

- Accession codes, unique identifiers, or web links for publicly available datasets
- A description of any restrictions on data availability
- For clinical datasets or third party data, please ensure that the statement adheres to our policy

Data for analyses of historical biogeography (trait-dependent and otherwise), as well as data related to the analysis of differences in the magnitude of trait effects

## Human research participants

Policy information about <u>studies involving human research participants and Sex and Gender in Research.</u>

| | |
|---|---|
| Reporting on sex and gender | N/A |
| Population characteristics | N/A |
| Recruitment | N/A |
| Ethics oversight | N/A |

Note that full information on the approval of the study protocol must also be provided in the manuscript.

# Field-specific reporting

Please select the one below that is the best fit for your research. If you are not sure, read the appropriate sections before making your selection.

☐ Life sciences    ☐ Behavioural & social sciences    ☒ Ecological, evolutionary & environmental sciences

For a reference copy of the document with all sections, see nature.com/documents/nr-reporting-summary-flat.pdf

# Ecological, evolutionary & environmental sciences study design

All studies must disclose on these points even when the disclosure is negative.

| | |
|---|---|
| Study description | We compiled phylogenetic, trait and species distribution data for 56 clades of tetrapods (spread across 10 amphibian clades, 15 mammal clades, 17 reptile clades, and 14 bird clades). We used trait-dependent and trait-independent biogeographic models to investigate the effect of traits in clades' biogeographic histories. We further analysed differences in the magnitude of trait effects and trait-dispersal patterns between clades. |
| Research sample | We compiled species-level trait data, species distribution data and phylogenetic data for 56 clades of tetrapods (7009 species spread across 10 amphibian clades, 15 mammal clades, 17 reptile clades, and 14 bird clades). These clades were chosen based on data availability. Trait data contained body size and life history traits. Species distribution data contained polygons of ranges and point data. Phylogenetic data was compiled in form of dated phylogenetic trees. Data sources are listed in the methods section of the main manuscript and in the extended data files. |
| Sampling strategy | N/A |
| Data collection | Data were collected from publicly available sources by the first author with help from the third author. |
| Timing and spatial scale | Data were compiled from publicly available sources between 04/2020 and 04/2021. |
| Data exclusions | No data were excluded. |
| Reproducibility | We have provided data and code to repeat our analyses. |
| Randomization | Randomization was not applicable since our study is not experimental. |
| Blinding | Blinding was not applicable since our study is not experimental. |

Did the study involve field work?    ☐ Yes    ☒ No

# Reporting for specific materials, systems and methods

We require information from authors about some types of materials, experimental systems and methods used in many studies. Here, indicate whether each material, system or method listed is relevant to your study. If you are not sure if a list item applies to your research, read the appropriate section before selecting a response.

## Materials & experimental systems

| n/a | Involved in the study |
|-----|----------------------|
| ☒ | Antibodies |
| ☒ | Eukaryotic cell lines |
| ☒ | Palaeontology and archaeology |
| ☒ | Animals and other organisms |
| ☒ | Clinical data |
| ☒ | Dual use research of concern |

## Methods

| n/a | Involved in the study |
|-----|----------------------|
| ☒ | ChIP-seq |
| ☒ | Flow cytometry |
| ☒ | MRI-based neuroimaging |

