## [Peer Review File · Nature Ecology & Evolution]

Peer Review Information

Journal: Nature Ecology & Evolution

Manuscript Title: Body size and life history shape the historical biogeography of tetrapods

Corresponding author name(s): Sarah-Sophie Weil

Editorial Notes:

Reviewer Comments & Decisions:

Decision Letter, initial version:

20th February 2023

Dear Dr Weil,

Your manuscript entitled "Uncovering how traits have shaped the historical biogeography of tetrapods" has now been seen by 2 reviewers, whose comments are attached. The reviewers have raised a number of concerns which will need to be addressed before we can offer publication in Nature Ecology & Evolution. We will therefore need to see your responses to the criticisms raised and to some editorial concerns, along with a revised manuscript, before we can reach a final decision regarding publication.

We therefore invite you to revise your manuscript taking into account all reviewer and editor comments. Please highlight all changes in the manuscript text file [OPTIONAL: in Microsoft Word format].

* If you have not done so already please begin to revise your manuscript so that it conforms to our Article format instructions at <http://www.nature.com/natecolevol/info/final-submission>. Refer also to any guidelines provided in this letter.

2[REDACTED]

Nature Ecology & Evolution is committed to improving transparency in authorship. As part of our efforts in this direction, we are now requesting that all authors identified as 'corresponding author' on published papers create and link their Open Researcher and Contributor Identifier (ORCID) with their account on the Manuscript Tracking System (MTS), prior to acceptance. ORCID helps the scientific community achieve unambiguous attribution of all scholarly contributions. You can create and link your ORCID from the home page of the MTS by clicking on 'Modify my Springer Nature account'. For more information please visit www.springernature.com/orcid.

[REDACTED]

Reviewer expertise:

Reviewer #1: Phylogenetic methods, including Phylogenetic Factor Analysis

Reviewer #2: Traits and biogeography, phylogenetics

Reviewers' comments:

Reviewer #1 (Remarks to the Author):

General comments: this manuscript presents a large-scale study to investigate how body size and life history strategy traits affect biogeographic dispersal in tetrapods. The authors demonstrate that including such traits in biogeographic models improves the explanation power, and they further examine which variables are correlated with the observed trait-dispersal relationships. To me this is a

2substantial work with three main contributions: (1) it provides new insights into how morphological and life-history traits shaped biogeography in tetrapods at a global scale (2) it stands as a successful application of cutting-edge biogeographic and phylogenetic methods (3) it collects an impressive data set and also develops a general workflow using biographic models, PFA, and regression analysis to assess the effect of traits on biogeography.

In general the manuscript is easy-to-follow. The data collection and pre-processing sections are thorough, and the result interpretation part is in-depth and informative. But certain parts are unclear or confusing and I hope my comments below can help the authors to improve this manuscript.

Major comments:

(1) Methods like MDMMs and the BioGeoBEARS models would benefit from a short description like those found in the previous work Weil et al. Without prior knowledge it is not clear what exact ecological/biological processes these models are estimating. Also, as incorporating traits in these models is an essential step, the readers may want to know more details.

(2) It is unclear how trait data were fit by the PFA model. Until line 72 "to determine the two traits of interests (body size and life history), we used phylogenetic factor analysis ...", I thought body size and life size history are two individual traits. But it seems PFA was used to separate the observed traits (like SVL, ES, SM .. under Methods - Trait data section) into multiple latent factors, where the first one is for body size and the rest for body size-independent traits. If so, it should be stated earlier that body size and life size history are not directly observed but inferred traits. Also, a brief, intuitive explanation on the latent factors and loadings would be necessary since they are cited multiple times in text and Extended Data Figure 1.

(3) Figure 2 can be improved. I only figured out the meaning of "extreme" in 2(b)(c) much later in the text. Also, which specific models were tested to get the AICc results in 2(c)?

(4) Why PFA is applied to all clades separately, instead of one PFA per major class (amphibians, mammals, birds, reptiles)?

(5) In the regression analysis to detect factors affecting the trait-dispersal links, are clades treated as independent of each other? If so, the authors should justify why ignoring the evolutionary relationship here is not a problem.

Minor things:

(1) I suggest changing wording like "better biogeographic models" to more specific statistical language, or adding some performance details when claiming "better models".

(2) AICc should be spelled out when it first appeared in the text.

Reviewer #2 (Remarks to the Author):

3The authors studied the relationship between two trait syndromes (body size and life history) and macroevolutionary/lineage dispersal in tetrapods. They used a large collection of available trait, distribution and phylogenetic data. They find that models including trait as a modifier of dispersal rate are supported in 91% of clades studied. They also found that the trait effects were themselves modulated by continental vs. oceanic disjunction type and the degree of trait variability. Large body size and fast life history facilitated dispersal in most cases, but in some cases, relationships were more complex, including cases where extreme or intermediate trait values favoured dispersal.

The study is original and significant in: (i) studying a key question of interest in biogeography, namely the effect of biotic traits on dispersal rates; and (ii) employing a large dataset and appropriate methods of analysis, with a few reservations/remarks I outline below, (iii) focusing on a group of organisms – tetrapods – with ecosystem importance and very good data coverage.

The authors employ state-of-the-art macroevolutionary and regression models. They also look at how the results could be affected by a range of confounding variables, such as phylogenetic uncertainty. The results and conclusions appear justified overall, with a few reservations outlined below. Sample sizes (number of phylogenies and clade size) and missing data were reported. I could not look at the data & scripts in the archive as the FigShare DOI provided (10.6084/m9.figshare.21897003) is not accessible. I appreciated that the authors reported the CO2eq emissions of their analyses (lines 594-596)!

The main conclusions are valid, although the authors overgeneralise a little, see suggestions below.

Some rewriting could help clarify the text in a few places but overall it is well written.

The references are appropriate; some suggestions given below.

I saw no inflammatory material nor do I have Diversity, Equity and Inclusion concerns.

I provide my review with the reservation that I have expertise in the macroevolutionary analyses, less so in the post-hoc regressions the authors used.

Suggestions:

General comments

The authors study macroevolutionary/lineage dispersal, but in several places in the introduction and conclusion, it is confounded with ecological/population dispersal. The distinction is important, because these two processes play out on very different time scales and the traits involved are not necessarily the same. E.g. in the abstract (line 12), the authors cite "disperser traits", but it is not clear whether traits favouring successful movement, establishment, or both, are meant. Line 224 suggests the authors are well aware of the difference between the types of dispersal ("we studied successful biogeographic dispersal"). In line 56, dispersal under global change is cited as a justification for the importance of the study, but the reference cited (ref. 14) deals with population dispersal, which is

4probably implicit here and thus different from the type of dispersal the authors studied. The last sentence of the conclusion (line 343) has the ambiguous term "biogeographic dispersal". The paper would benefit from clearly stating the differences between the two types of dispersal in the introduction and defining what is studied here; some published reviews (Gillespie et al. 2012 [cited, ref. 3]; Ricklefs & Jenkins et al. 2011, doi.org/10.1098/rstb.2011.0066; Hackel & Sanmartin 2021, doi.org/10.1016/j.tree.2021.07.007) might be useful here.

There is some overgeneralisation which should be revisited. In the title, "traits" is too generic, replacing it by "body size and life history" would be appropriate. In the conclusion (line 320), "morphological" should be replaced with "body size". Also in the conclusion, the authors claim "we find that species' traits have played a key role in shaping global biogeography and the diversity of life on Earth" (line 323). This should be toned down; with all due respect for this large analysis and the effort involved, the authors have only studied two trait syndromes and one (comparatively small) small clade of life on earth.

I miss some awareness and discussion of phylogenetic scale: The crown divergences of the different clades studied probably differ quite a lot – how could this have affected the analysis? Don't we expect biogeographic signal (i.e. dispersals reconstructed) to be stronger in younger clades? At the least, I would like to see crown ages given (perhaps in the Fig. 1) overview; maybe they could even be included as control variables in the downstream analyses. The review of Graham et al. 2018 (doi.org/10.1111/geb.12686) could be a useful guide.

The authors have made some laudable effort to account for phylogenetic uncertainty. The impact in Natricinae on the conclusions is somewhat concerning but I do understand the computational limitations. Could one rough approach to account for uncertainty be the inclusion of average node support as control variable in the downstream analyses?

I also wonder about the risk of inferring spurious trait effects (type I error) simply because any free parameter added generally improves model fit in macroevolution. Especially as line 104 says "The number of estimated dispersal events per clade did not differ significantly between trait-dependent and trait-independent models" – even though adding a trait effect as parameter improves model fit. Could this issue at least be discussed? This concern has driven the development of hidden-state macroevolutionary models, with some recent application in biogeography (see Caetano et al. 2018, doi.org/10.1111/evo.13602).

Another general problem in studying macroevolution is extinction: I would like to see at least some discussion of how extinction could have impacted the results – the models used also estimate an extinction parameter, e . Are traits that favour dispersal also traits that reduce extinction risk, or could there be more complex scenarios with tradeoffs?

A final suggestion: It may be useful for the reader to translate the parameters and effect sizes estimated (i.e. dispersal rate) into real-world numbers in the discussion. E.g., how many more dispersal events over the X million years of history in a clade can we attribute to the effect of a trait?

Specific comments

516-19 - long, convolute sentence, split in two?

Lines 29-48 - it should be made clear here what applies to tetrapods vs. animals or all organisms. Eg. line 32 ("Short-distance dispersal is primarily determined by active movements") certainly does not apply to plants!

Line 47, "general trait-dispersal bias" – not clear to me what bias is meant here, can you reword?

Line 113 - Why is the small number of nodes "significant" here? As the following sentence seems quite vague to me, I wonder if this part is really necessary.

Lines 147, 232, and probably other places - the use of "class" is sometimes confusing, as one could think of the trait bins/classes. Maybe use "tetrapod class" throughout for clarity?

537 and following: it would be useful here for the reader to name the parameter of interest (dispersal rate – d ?) that is estimated in DEC and the other models.

Line 561: I see two problems with this first model selection step: (1) Model selection is problematic here because these models do not differ in their free parameters, only in the events allowed at nodes (see Ree & Sanmartín, doi.org/10.1111/jbi.13173 – note this is not only a problem of DEC vs. DEC+J). (2) The models differ in the types of vicariance allowed at a node. This could impact the numbers and rates of dispersals inferred and thus introduce another source of variation between clades. One solution could be to just choose one model for all clades, another to include the type of model as control/random variable in the downstream analyses.

597 - The effect - "of" missing

Fig. 1 - The small inset phylogenies and the maps are not really helpful, too much information crammed in. As this is only an overview, maybe just showing colour codes for distributions (e.g. coloured squares for continents/major realms), and potential age and no. of species, would be more useful.

Fig. 2 - a common colour legend would be helpful. For 2a the reader has to refer to the caption.

Fig 2c - it would be helpful to highlight the 50% line.

Fig. 2d - check if the symbols for rows and columns make sense – according to the legend, they indicate combinations of body size and life history.

Fig 3a - could the figure size be increased? Lots of white space.

Extended data table 1 is missing, I only see the caption.

Extended table 3 - check number of clades (last column): they are the same for both inclusion

thresholds, shouldn't they be lower for the higher threshold (20%)?

*****END*****

Author Rebuttal to Initial comments

Reviewers' comments:

Reviewer #1 (Remarks to the Author):

General comments: this manuscript presents a large-scale study to investigate how body size and life history strategy traits affect biogeographic dispersal in tetrapods. The authors demonstrate that including such traits in biogeographic models improves the explanation power, and they further examine which variables are correlated with the observed trait-dispersal relationships. To me this is a substantial work with three main contributions: (1) it provides new insights into how morphological and life-history traits shaped biogeography in tetrapods at a global scale (2) it stands as a successful application of cutting-edge biogeographic and phylogenetic methods (3) it collects an impressive data set and also develops a general workflow using biogeographic models, PFA, and regression analysis to assess the effect of traits on biogeography.

In general the manuscript is easy-to-follow. The data collection and pre-processing sections are thorough, and the result interpretation part is in-depth and informative. But certain parts are unclear or confusing and I hope my comments below can help the authors to improve this manuscript.

Thank you for this positive and detailed assessment of our manuscript. We hope that we have now clarified the confusing parts.

Major comments:

7(1) Methods like MDMMs and the BioGeoBEARS models would benefit from a short description like those found in the previous work Weil et al. Without prior knowledge it is not clear what exact ecological/biological processes these models are estimating. Also, as incorporating traits in these models is an essential step, the readers may want to know more details.

We now provide a more detailed description of biogeographic models (lines 722-726 and lines 748-755, pages 31-32). A detailed description of the implementation of trait-dependency is available in the Methods section "Trait-dispersal relationships". We have cross-referenced this section in the new section in lines 764-766, page 32.

(2) It is unclear how trait data were fit by the PFA model. Until line 72 "to determine the two traits of interests (body size and life history), we used phylogenetic factor analysis ...", I thought body size and life size history are two individual traits. But it seems PFA was used to separate the observed traits (like SVL, ES, SM .. under Methods - Trait data section) into multiple latent factors, where the first one is for body size and the rest for body size-independent traits. If so, it should be stated earlier that body size and life size history are not directly observed but inferred traits. Also, a brief, intuitive explanation on the latent factors and loadings would be necessary since they are cited multiple times in text and Extended Data Figure 1.

In the introduction we have clarified this point by adding "body size-independent life history strategy" and "inferred through phylogenetic factor analysis" in lines 65-66, page 3. More detail was also added further in the text (lines 98-103, page 5) where we describe the use of phylogenetic factor analysis:

"To determine the two traits of interest (body size and life history strategy), we used phylogenetic factor analysis^{25,26} to position species along two main latent factors of trait variation per clade. The first factor represented body size and related life history trait covariation, and the second factor body size independent life history covariation. For most clades, the life history factor was consistent with a fast-slow life history continuum, mainly determined by body size-independent clutch/litter size (Extended Data Fig. 1)."

(3) Figure 2 can be improved. I only figured out the meaning of “extreme” in 2(b)(c) much later in the text. Also, which specific models were tested to get the AICc results in 2(c)?

We have added a sentence in the Figure legend to clarify the confusion around species with “extreme” traits: “Extreme body size/life history” refers to clades where a u-shaped relationship was inferred between traits and dispersal rates, i.e. where species with extremely small or large body sizes, or fast or slow life histories had a dispersal advantage.” We have also added more detail in the legend regarding the models that were tested to get the AICc weights in 2c (the sum of AICc weights of trait-dependent models (+m2 and +m2x versions, see Methods) compared to trait-independent models (base model and +x version), averaged across four binary thresholds).

(4) Why PFA is applied to all clades separately, instead of one PFA per major class (amphibians, mammals, birds, reptiles)?

It is known that there is a multitude of trade-offs structuring species’ life histories (Stearns 1992), and these trade-offs vary across clades (we indeed found that the fast-slow continuum structured life history in 47/56 clades). Had we conducted PFAs at a higher taxonomic level, there would have been a risk that trade-offs of species-rich clades determine the loadings of the trait variables on the two extracted factors. Thus, trade-offs playing a role at clade-level could have been concealed and species’ might have been ranked inappropriately. We have added this explanation in lines 785-787, page 33: “We analysed all clades separately as trade-offs structuring species’ life histories can vary between clades and biogeographic models were conducted at clade-level.”.

(5) In the regression analysis to detect factors affecting the trait-dispersal links, are clades treated as independent of each other? If so, the authors should justify why ignoring the evolutionary relationship here is not a problem.

Clades are indeed treated as independent of each other in the regressions, as when we tested for phylogenetic signal in the residuals of the models (as suggested by Revell 2012) we found none. We have clarified this point with the following sentence in the Methods section (lines 902-904, page 37): “There

was no phylogenetic signal in the residuals of any of the models (function 'phylosig', R package phytools v1.2-0¹⁸), therefore there was no need to account for phylogenetic relationships between clades."

Minor things:

(1) I suggest changing wording like "better biogeographic models" to more specific statistical language, or adding some performance details when claiming "better models".

We have reformulated in the abstract: "Biogeographic models incorporating body size or life history accrued more statistical support than trait-independent models in 91% of clades." (lines 10-12). Furthermore, we have clarified in lines 211-212, page 9 what we mean by "better biogeographic models":

"(i.e. the best trait-dependent model accrued more than 50% of the AICc weight)".

(2) AICc should be spelled out when it first appeared in the text.

Done: "To assess whether the chosen traits have played a role in clades' biogeographic histories we compared the performance of trait-dependent and trait-independent models using their model weights based on the corrected Akaike Information Criterion (AICc)." (lines 132-134, page 6).

References:

Revell, L. J. (2010). Phylogenetic signal and linear regression on species data. Methods in Ecology and Evolution, 1(4), 319-329.

Stearns, S. C. (1992). The evolution of life histories. Oxford, United Kingdom:

10Oxford University Press.

Reviewer #2 (Remarks to the Author):

The authors studied the relationship between two trait syndromes (body size and life history) and macroevolutionary/lineage dispersal in tetrapods. They used a large collection of available trait, distribution and phylogenetic data. They find that models including trait as a modifier of dispersal rate are supported in 91% of clades studied. They also found that the trait effects were themselves modulated by continental vs. oceanic disjunction type and the degree of trait variability. Large body size and fast life history facilitated dispersal in most cases, but in some cases, relationships were more complex, including cases where extreme or intermediate trait values favoured dispersal.

The study is original and significant in: (i) studying a key question of interest in biogeography, namely the effect of biotic traits on dispersal rates; and (ii) employing a large dataset and appropriate methods of analysis, with a few reservations/remarks I outline below, (iii) focusing on a group of organisms – tetrapods – with ecosystem importance and very good data coverage.

The authors employ state-of-the-art macroevolutionary and regression models. They also look at how the results could be affected by a range of confounding variables, such as phylogenetic uncertainty. The results and conclusions appear justified overall, with a few reservations outlined below. Sample sizes (number of phylogenies and clade size) and missing data were reported. I could not look at the data & scripts in the archive as the FigShare DOI provided (10.6084/m9.figshare.21897003) is not accessible. I appreciated that the authors reported the CO₂e emissions of their analyses (lines 594-596)!

We apologise for not including the appropriate link to the data and scripts. Indeed, the given link will only become active upon publication. For review purposes, the following link can be used: <https://figshare.com/s/4b6cff9ac55dbe78425c>.

The main conclusions are valid, although the authors overgeneralise a little, see suggestions below.

Thank you for this assessment, we appreciate your suggestions and have edited the parts you indicated below to be more specific.

Some rewriting could help clarify the text in a few places but overall it is well written.

We have clarified the text in several instances according to your suggestions.

The references are appropriate; some suggestions given below.

We have included most of the references you suggested below.

I saw no inflammatory material nor do I have Diversity, Equity and Inclusion concerns.

I provide my review with the reservation that I have expertise in the macroevolutionary analyses, less so in the post-hoc regressions the authors used.

Suggestions:

General comments

The authors study macroevolutionary/lineage dispersal, but in several places in the introduction and conclusion, it is confounded with ecological/population dispersal. The distinction is important, because these two processes play out on very different time scales and the traits involved are not necessarily the

same. E.g. in the abstract (line 12), the authors cite "disperser traits", but it is not clear whether traits favouring successful movement, establishment, or both, are meant. Line 224 suggests the authors are well aware of the difference between the types of dispersal ("we studied successful biogeographic dispersal"). In line 56, dispersal under global change is cited as a justification for the importance of the study, but the reference cited (ref. 14) deals with population dispersal, which is probably implicit here and thus different from the type of dispersal the authors studied. The last sentence of the conclusion (line 343) has the ambiguous term "biogeographic dispersal". The paper would benefit from clearly stating the differences between the two types of dispersal in the introduction and defining what is studied here; some published reviews (Gillespie et al. 2012 [cited, ref. 3]; Ricklefs & Jenkins et al. 2011, doi.org/10.1098/rstb.2011.0066; Hackel & Sanmartín 2021, doi.org/10.1016/j.tree.2021.07.007) might be useful here.

Thank you for raising this issue and proposing relevant literature, these points can indeed lead to confusion. We now state at the end of the introduction (lines 67-69, page 3): "We thus focus on biogeographic dispersal (synonymous to "lineage dispersal" sensu Hackel & Sanmartín^{new22}) which includes both successful movement or transport to a new biogeographic region and establishment there.". In addition, we have replaced „disperser traits" in the abstract with "traits favouring successful biogeographic dispersal". We have also replaced reference 14 in line 74, page 3 with a more appropriate one (Estrada et al 2013) which focuses on the role of traits in species' range shifts. We have replaced "biogeographic dispersal" in the last sentence of the conclusion (line 479, page 20) with "successful dispersal across major biogeographic barriers".

There is some overgeneralisation which should be revisited. In the title, "traits" is too generic, replacing it by "body size and life history" would be appropriate. In the conclusion (line 320), "morphological" should be replaced with "body size". Also in the conclusion, the authors claim "we find that species' traits have played a key role in shaping global biogeography and the diversity of life on Earth" (line 323). This should be toned down; with all due respect for this large analysis and the effort involved, the authors have only studied two trait syndromes and one (comparatively small) small clade of life on earth.

We agree that including "body size and life history" is more appropriate in the title and have changed it accordingly. We also have replaced "morphological" by "body size" in the conclusion. In the conclusion (previous lines 320-323, new lines 440-442, page 19), we have rewritten the sentences as follows: "To

conclude, we show here for the first time that body size and life-history traits are related to, and interact in their influence on, biogeographic dispersal in a large number of tetrapod clades. These traits have thus played a key role in shaping global biogeography of tetrapods.”

I miss some awareness and discussion of phylogenetic scale: The crown divergences of the different clades studied probably differ quite a lot – how could this have affected the analysis? Don't we expect biogeographic signal (i.e. dispersals reconstructed) to be stronger in younger clades? At the least, I would like to see crown ages given (perhaps in the Fig. 1) overview; maybe they could even be included as control variables in the downstream analyses. The review of Graham et al. 2018 (doi.org/10.1111/geb.12686) could be a useful guide.

See response to the next comment/suggestion.

The authors have made some laudable effort to account for phylogenetic uncertainty. The impact in *Natricinae* on the conclusions is somewhat concerning but I do understand the computational limitations. Could one rough approach to account for uncertainty be the inclusion of average node support as control variable in the downstream analyses?

Thank you for these excellent suggestions. We have included crown age and average node resolution in the analysis of differences in the magnitude of trait effects between clades (lines 240-252, pages 10-11; Tab. 1, where we now also cite Graham et al 2018). Crown age was not related to trait effects likely because we fixed some root constraints in our analyses based on literature and larger-scale biogeographic studies (Extended Tab. 1) to avoid the potentially biased estimations for the deeper nodes of the phylogeny (Ronquist & Sanmartin 2011).

We have further modified Fig. 1 so that crown ages of clades can be placed on the timeline.

Node resolution was indeed related to the effect magnitude of life history in dispersal, likely because it was correlated with the number of dispersal events (which we used as a proxy of barrier strength). This correlation makes sense as increasing the number of movements in biogeographic histories is likely to make inferences by the model more difficult and thus increase estimation uncertainties (which we now mention in lines 932-935, page 38). Nevertheless, when added to the models node resolution was not selected in the step-selection process.

I also wonder about the risk of inferring spurious trait effects (type I error) simply because any free parameter added generally improves model fit in macroevolution. Especially as line 104 says "The number of estimated dispersal events per clade did not differ significantly between trait-dependent and trait-independent models" – even though adding a trait effect as parameter improves model fit. Could this issue at least be discussed? This concern has driven the development of hidden-state macroevolutionary models, with some recent application in biogeography (see Caetano et al. 2018, doi.org/10.1111/evo.13602).

This is a valid concern, and we are grateful for the opportunity to address it in lines 212-222, pages 9-10: "However, models with more free parameters have been shown to be more likely to be selected in model comparison solely due to their complexity, which can lead to misleading parameter estimates^{new30}. A solution for this problem consists in combining parameter estimates of several models, weighting them by their likelihoods penalized by the number of parameters of each model (e.g. using AICc)^{new31}. For each clade, we thus combined the dispersal parameters of all models via AICc-weighted averages to understand the role of body size and life history in dispersal." (new ref 30: Rabosky & Goldberg 2015, new ref 31: Caetano et al. 2018).

In addition, in our study we used several binarization thresholds per trait, and then averaged these results along the trait gradients. This approach is rather conservative in terms of detecting the effects of traits, and should therefore reduce Type 1 errors. If spurious trait effects were inferred in individual models, the overall effect of traits in the majority of the averaged trait-dispersal curves should have been negligible (which was not the case: 79% of the 56 body size-dispersal relationships and 83% of the 47 life history-dispersal relationships in which fast-slow consistent trade-offs structured life history showed a difference in estimated dispersal rates between trait states greater than 10%).

That the number of estimated dispersal events did not differ significantly between trait-dependent and trait-independent models is in itself also not necessarily a cause for concern. If dispersal is estimated to be higher for certain trait states, this might be compensated for by lower dispersal rates in the opposite trait states which would lead to no difference in the number of dispersal events overall. Indeed, we found that the difference between trait-dependent and trait-independent models might lie rather in a changed dispersal path and/or timing of dispersal events, instead of the total number of dispersal events (lines 166-167, page 7).

Another general problem in studying macroevolution is extinction: I would like to see at least some discussion of how extinction could have impacted the results – the models used also estimate an extinction parameter, e . Are traits that favour dispersal also traits that reduce extinction risk, or could there be more complex scenarios with tradeoffs?

This comment identifies two separate issues relating to two ways the term “extinction” is used in macroevolutionary and biogeographic studies. The first is species extinction which is famously difficult to infer from molecular phylogenies (Rabosky 2010). The most likely impact of species extinction on our results would be an erroneous estimation of past range and trait evolution due to missing tips. This could also impact the inferred trait-dispersal relationships. A partial remedy to this problem might be to include fossils (which is possible in BioGeoBEARS) or to use recently developed models that model extinct branches explicitly (Herrera-Alsina et al. 2022). While including these models were impractical for our study (due to computational limitations and the missing possibility to include traits), we now develop this idea in lines 454-459, page 19.

The second way “extinction” is used concerns the parameter “ e ” in BioGeoBEARS models. It is correct that this parameter is called “extinction parameter”, however, it refers to range contraction (local extinction in a specific region) and not complete extinction of the lineage (we have clarified this in lines 749, page 32). In BioGeoBEARS models as they currently exist it is not possible to determine if traits favouring dispersal reduce risk of range contraction. However, this would be quite straightforward to implement (analogous to traits modifying dispersal rates, local extinction rates could be modified as well). We agree with you that it would be very valuable to assess the effect of traits on both dispersal and range contraction; together this might be a good indicator of extinction risk (e.g. if certain traits for example are related simultaneously to low dispersal rates and high range contraction rates). We have added this idea in lines 460-471, pages 19-20.

A final suggestion: It may be useful for the reader to translate the parameters and effect sizes estimated (i.e. dispersal rate) into real-world numbers in the discussion. E.g., how many more dispersal events over the X million years of history in a clade can we attribute to the effect of a trait?

We have added a sentence in lines 228-213, page 10: “This corresponds to a median increase in dispersal rates of lineages with disperser traits of ca. 0.02 dispersal events/million years (standard deviation body size: 0.06; life history: 0.13), compared to dispersal rates of lineages with non-disperser traits”.

Specific comments

16-19 - long, convolute sentence, split in two?

Done.

Lines 29-48 - it should be made clear here what applies to tetrapods vs. animals or all organisms. Eg. line 32 ("Short-distance dispersal is primarily determined by active movements") certainly does not apply to plants!

We have added "in animals" in two instances in this paragraph to clarify this.

Line 47, "general trait-dispersal bias" – not clear to me what bias is meant here, can you reword?

We have rephrased the sentence as follows: "However, we do not know whether these initial findings in three small reptile groups represent a general pattern of trait-dispersal relationships in tetrapods, or whether multiple relationships exist across highly different clades."

Line 113 - Why is the small number of nodes "significant" here? As the following sentence seems quite vague to me, I wonder if this part is really necessary.

Indeed, the word "significant" is inappropriate here, we have reformulated the sentence as follows: "However, in a small number of nodes (6% on average), the identity of ancestral ranges differs between trait-dependent and trait-independent estimations." (lines 165-166, page 7).

Lines 147, 232, and probably other places - the use of "class" is sometimes confusing, as one could think of the trait bins/classes. Maybe use "tetrapod class" throughout for clarity?

Thank you for this suggestion, we have modified the text accordingly.

537 and following: it would be useful here for the reader to name the parameter of interest (dispersal rate – d ?) that is estimated in DEC and the other models.

We have included a more detailed description of biogeographic models in lines 723-726 and lines 748-755, pages 31-32.

Line 561: I see two problems with this first model selection step: (1) Model selection is problematic here because these models do not differ in their free parameters, only in the events allowed at nodes (see Ree & Sanmartín, doi.org/10.1111/jbi.13173 – note this is not only a problem of DEC vs. DEC+J). (2) The models differ in the types of vicariance allowed at a node. This could impact the numbers and rates of dispersals inferred and thus introduce another source of variation between clades. One solution could be to just choose one model for all clades, another to include the type of model as control/random variable in the downstream analyses.

Thank you for raising this issue, we have included the category of base model used for the biogeographic estimations in the analysis of differences in the magnitude of trait effects between clades (lines 252, page 11; Tab. 1). However, it did not impact the magnitude of trait effects in past dispersal.

597 - The effect - "of" missing

Thanks for catching this.

Fig. 1 - The small inset phylogenies and the maps are not really helpful, too much information crammed in. As this is only an overview, maybe just showing colour codes for distributions (e.g. coloured squares for continents/major realms), and potential age and no. of species, would be more useful.

We have reworked the figure according to your suggestions.

Fig. 2 - a common colour legend would be helpful. For 2a the reader has to refer to the caption.

We have added “body size” and “life history” in 2a to clarify this. Everything blue is related to body size and everything yellow to life history.

Fig 2c - it would be helpful to highlight the 50% line.

Done.

Fig. 2d - check if the symbols for rows and columns make sense – according to the legend, they indicate combinations of body size and life history.

This is correct, the ‘combinations’ in the legend refers to the colour of the squares, not the symbols. We have added a sentence in the figure legend to clarify this panel: “The symbols are the same as in c), indicating the shape of the relationships between traits (body size and life history) and dispersal rates.”.

Fig 3a - could the figure size be increased? Lots of white space.

Done.

Extended data table 1 is missing, I only see the caption.

This table was provided as a separate .csv table as it was too large to be included directly. It should have been provided with the other supplementary information file.

Extended table 3 - check number of clades (last column): they are the same for both inclusion thresholds, shouldn't they be lower for the higher threshold (20%)?

The numbers in this last column are correct. We included a column "Identified relationships - NONE" - relationships where the difference was less than 10% and 20%, respectively, are counted towards this column.

References:

*Estrada, A., Morales-Castilla, I., Caplat, P., & Early, R. (2016). Usefulness of species traits in predicting range shifts. *Trends in Ecology & Evolution*, 31(3), 190-203.*

*Herrera-Alsina, L., Algar, A. C., Lancaster, L. T., Ornelas, J. F., Bocedi, G., Papadopoulos, A. S., ... & Travis, J. M. (2022). The missing link in biogeographic reconstruction: Accounting for lineage extinction rewrites history. *Journal of Biogeography*, 49(11), 1941-1951.*

*Rabosky, D. L. (2010). "Extinction rates should not be estimated from molecular phylogenies". *Evolution* 64, 6, 1816–1824.*

*Ronquist, F., & Sanmartín, I. (2011). Phylogenetic methods in biogeography. *Annual Review of Ecology, Evolution, and Systematics*, 42, 441-464.*

*****END*****

Decision Letter, first revision:

17th May 2023

Dear Dr. Weil,

Thank you for submitting your revised manuscript "Uncovering how body size and life history have shaped the historical biogeography of tetrapods" (NATECOLEVOL-23010104A). It has now been seen again by the original reviewers and their comments are below. The reviewers find that the paper has improved in revision, and therefore we'll be happy in principle to publish it in Nature Ecology & Evolution, pending minor revisions to satisfy the reviewers' final requests and to comply with our editorial and formatting guidelines.

[REDACTED]

Reviewer #1 (Remarks to the Author):

I think the authors have effectively addressed my previous comments, resulting in significant improvements to the manuscript.

I just have one minor suggestion: in Extended Data Figure 1, the x-axis labels can be rotated so it's easier to read. It may be sufficient to include these labels only in subfigure (e) and (f).

Reviewer #2 (Remarks to the Author):

21I am satisfied with the authors' revision and would like to congratulate them on a very interesting study that will certainly move the field forward.

Two minor remarks:

- the line numbers in the rebuttal did not match with the manuscript so it took some searching to locate the modified parts. I don't know if this is just a mistake that crept in or if it's related to the manuscript submission system somehow added new line numbers.

- Line 154, "models with more free parameters have been shown to be more likely to be selected in model comparison": I appreciate that the authors now discuss the issue of model complexity. It would be helpful to clarify that this is a specific concern for `_macroevolutionary_` models. AIC already corrects for model complexity, but the problem in macroevolution is that there are many potential sources of variation and our models are still so simple that any trait/effect added tends to increase model fit even with AIC - statistically correct but biologically potentially misleading.

signed - Jan Hackel

Our ref: NATECOLEVOL-23010104A

8th June 2023

Dear Dr. Weil,

Thank you for your patience as we've prepared the guidelines for final submission of your Nature Ecology & Evolution manuscript, "Uncovering how body size and life history have shaped the historical biogeography of tetrapods" (NATECOLEVOL-23010104A). Please carefully follow the step-by-step instructions provided in the attached file, and add a response in each row of the table to indicate the changes that you have made. Please also check and comment on any additional marked-up edits we have proposed within the text. Ensuring that each point is addressed will help to ensure that your revised manuscript can be swiftly handed over to our production team.

**We would like to start working on your revised paper, with all of the requested files and forms, as soon as possible (preferably within two weeks). Please get in contact with us immediately if you

22anticipate it taking more than two weeks to submit these revised files.**

In recognition of the time and expertise our reviewers provide to Nature Ecology & Evolution's editorial process, we would like to formally acknowledge their contribution to the external peer review of your manuscript entitled "Uncovering how body size and life history have shaped the historical biogeography of tetrapods". For those reviewers who give their assent, we will be publishing their names alongside the published article.

Nature Ecology & Evolution offers a Transparent Peer Review option for new original research manuscripts submitted after December 1st, 2019. As part of this initiative, we encourage our authors to support increased transparency into the peer review process by agreeing to have the reviewer comments, author rebuttal letters, and editorial decision letters published as a Supplementary item. When you submit your final files please clearly state in your cover letter whether or not you would like to participate in this initiative. Please note that failure to state your preference will result in delays in accepting your manuscript for publication.

Cover suggestions

As you prepare your final files we encourage you to consider whether you have any images or illustrations that may be appropriate for use on the cover of Nature Ecology & Evolution.

Nature Ecology & Evolution has now transitioned to a unified Rights Collection system which will allow our Author Services team to quickly and easily collect the rights and permissions required to publish

23your work. Approximately 10 days after your paper is formally accepted, you will receive an email in providing you with a link to complete the grant of rights. If your paper is eligible for Open Access, our Author Services team will also be in touch regarding any additional information that may be required to arrange payment for your article.

Please note that *Nature Ecology & Evolution* is a Transformative Journal (TJ). Authors may publish their research with us through the traditional subscription access route or make their paper immediately open access through payment of an article-processing charge (APC). Authors will not be required to make a final decision about access to their article until it has been accepted. [Find out more about Transformative Journals](https://www.springernature.com/gp/open-research/transformative-journals)

Authors may need to take specific actions to achieve [compliance with funder and institutional open access mandates](https://www.springernature.com/gp/open-research/funding/policy-compliance-faqs). If your research is supported by a funder that requires immediate open access (e.g. according to [Plan S principles](https://www.springernature.com/gp/open-research/plan-s-compliance)) then you should select the gold OA route, and we will direct you to the compliant route where possible. For authors selecting the subscription publication route, the journal's standard licensing terms will need to be accepted, including [self-archiving-and-license-to-publish](https://www.nature.com/nature-portfolio/editorial-policies/self-archiving-and-license-to-publish). Those licensing terms will supersede any other terms that the author or any third party may assert apply to any version of the manuscript.

[REDACTED]

[REDACTED]

Reviewer #1:

Remarks to the Author:

I think the authors have effectively addressed my previous comments, resulting in significant

24improvements to the manuscript.

I just have one minor suggestion: in Extended Data Figure 1, the x-axis labels can be rotated so it's easier to read. It may be sufficient to include these labels only in subfigure (e) and (f).

Reviewer #2:

Remarks to the Author:

I am satisfied with the authors' revision and would like to congratulate them on a very interesting study that will certainly move the field forward.

Two minor remarks:

- the line numbers in the rebuttal did not match with the manuscript so it took some searching to locate the modified parts. I don't know if this is just a mistake that crept in or if it's related to the manuscript submission system somehow added new line numbers.

- Line 154, "models with more free parameters have been shown to be more likely to be selected in model comparison": I appreciate that the authors now discuss the issue of model complexity. It would be helpful to clarify that this is a specific concern for *macroevolutionary* models. AIC already corrects for model complexity, but the problem in macroevolution is that there are many potential sources of variation and our models are still so simple that any trait/effect added tends to increase model fit even with AIC - statistically correct but biologically potentially misleading.

signed - Jan Hackel

Author Rebuttal, first revision:

Please find below our response to the remaining reviewer comments. Our responses are in blue italics.

Reviewer #1:

Remarks to the Author:

I think the authors have effectively addressed my previous comments, resulting in significant improvements to the manuscript.

I just have one minor suggestion: in Extended Data Figure 1, the x-axis labels can be rotated so it's easier to read. It may be sufficient to include these labels only in subfigure (e) and (f).

RESPONSE: Thank you for this assessment and helping us improve the manuscript. We have edited Extended Data Figure 1 as suggested.

Reviewer #2:

Remarks to the Author:

I am satisfied with the authors' revision and would like to congratulate them on a very interesting study that will certainly move the field forward.

Two minor remarks:

- the line numbers in the rebuttal did not match with the manuscript so it took some searching to locate the modified parts. I don't know if this is just a mistake that crept in or if it's related to the manuscript submission system somehow added new line numbers.

RESPONSE: We apologize for this. Unfortunately, it is a recurrent problem in Microsoft Word that track changes cause line numbers to skip between pages (see here for example: <https://answers.microsoft.com/en-us/msoffice/forum/all/complaint-tracked-changes-causes-line-numbers-to/5d76fb35-8686-4e17-abe2-d85f2ef06b5e>).

- Line 154, "models with more free parameters have been shown to be more likely to be selected in model comparison": I appreciate that the authors now discuss the issue of model complexity. It would be helpful to clarify that this is a specific concern for `_macroevolutionary_` models. AIC already corrects for model complexity, but the problem in macroevolution is that there are many potential sources of variation and our models are still so simple that any trait/effect added tends to increase model fit even with AIC - statistically correct but biologically potentially misleading.

RESPONSE: Thank you for this clarification, we have added "macroevolutionary" in the mentioned sentence.

signed - Jan Hackel

Final Decision Letter:

4th July 2023

Dear Dr Weil,

We are pleased to inform you that your Article entitled "Body size and life history shape the historical biogeography of tetrapods", has now been accepted for publication in Nature Ecology & Evolution.

Over the next few weeks, your paper will be copyedited to ensure that it conforms to Nature Ecology and Evolution style. Once your paper is typeset, you will receive an email with a link to choose the appropriate publishing options for your paper and our Author Services team will be in touch regarding any additional information that may be required

You will not receive your proofs until the publishing agreement has been received through our system

Due to the importance of these deadlines, we ask you please us know now whether you will be difficult to contact over the next month. If this is the case, we ask you provide us with the contact information (email, phone and fax) of someone who will be able to check the proofs on your behalf, and who will be available to address any last-minute problems . Once your paper has been scheduled for online publication, the Nature press office will be in touch to confirm the details.

Acceptance of your manuscript is conditional on all authors' agreement with our publication policies (see www.nature.com/authors/policies/index.html). In particular your manuscript must not be published elsewhere and there must be no announcement of the work to any media outlet until the publication date (the day on which it is uploaded onto our web site).

Please note that *Nature Ecology & Evolution* is a Transformative Journal (TJ). Authors may publish their research with us through the traditional subscription access route or make their paper immediately open access through payment of an article-processing charge (APC). Authors will not be required to make a final decision about access to their article until it has been accepted. [Find out more about Transformative Journals](https://www.springernature.com/gp/open-research/transformative-journals)

Authors may need to take specific actions to achieve [compliance](https://www.springernature.com/gp/open-research/funding/policy-compliance-faqs) with funder and institutional open access mandates. If your research is supported by a funder that requires immediate open access (e.g. according to [Plan S principles](https://www.springernature.com/gp/open-research/plan-s-compliance)) then you should select the gold OA route, and we will direct you to the compliant route where

27possible. For authors selecting the subscription publication route, the journal's standard licensing terms will need to be accepted, including <https://www.nature.com/nature-portfolio/editorial-policies/self-archiving-and-license-to-publish>. Those licensing terms will supersede any other terms that the author or any third party may assert apply to any version of the manuscript.

We welcome the submission of potential cover material (including a short caption of around 40 words) related to your manuscript; suggestions should be sent to Nature Ecology & Evolution as electronic files (the image should be 300 dpi at 210 x 297 mm in either TIFF or JPEG format). Please note that such pictures should be selected more for their aesthetic appeal than for their scientific content, and that colour images work better than black and white or grayscale images. Please do not try to design a cover with the Nature Ecology & Evolution logo etc., and please do not submit composites of images related to your work. I am sure you will understand that we cannot make any promise as to whether any of your suggestions might be selected for the cover of the journal.

You can generate the link yourself when you receive your article DOI by entering it here: <http://authors.springernature.com/share>.

Yours sincerely,

Simon Harold PhD
Senior Editor

28Nature Ecology and Evolution

P.S. Click on the following link if you would like to recommend Nature Ecology & Evolution to your librarian <http://www.nature.com/subscriptions/recommend.html#forms>

** Visit the Springer Nature Editorial and Publishing website at http://editorial-jobs.springernature.com?utm_source=ejp_NEcoE_email&utm_medium=ejp_NEcoE_email&utm_campaign=ejp_NEcoE for more information about our career opportunities. If you have any questions please click [here](mailto:editorial.publishing.jobs@springernature.com). **